# Glutaredoxin catalysis requires two distinct glutathione interaction sites

Patricia Begas[1],*, Linda Liedgens[1],*, Anna Moseler[2], Andreas J. Meyer[2] & Marcel Deponte[1]

Glutaredoxins are key players in cellular redox homoeostasis and exert a variety of essential functions ranging from glutathione-dependent catalysis to iron metabolism. The exact structure–function relationships and mechanistic differences among glutaredoxins that are active or inactive in standard enzyme assays have so far remained elusive despite numerous kinetic and structural studies. Here, we elucidate the enzymatic mechanism showing that glutaredoxins require two distinct glutathione interaction sites for efficient redox catalysis. The first site interacts with the glutathione moiety of glutathionylated disulfide substrates. The second site activates glutathione as the reducing agent. We propose that the requirement of two distinct glutathione interaction sites for the efficient reduction of glutathionylated disulfide substrates explains the deviating structure–function relationships, activities and substrate preferences of different glutaredoxin subfamilies as well as thioredoxins. Our model also provides crucial insights for the design or optimization of artificial glutaredoxins, transition-state inhibitors and glutaredoxin-coupled redox sensors.

[1] Department of Parasitology, Ruprecht-Karls University, Im Neuenheimer Feld 324, D-69120 Heidelberg, Germany. [2] Institute of Crop Science and Resource Conservation (INRES)-Chemical Signalling, University of Bonn, D-53113 Bonn, Germany. * These authors contributed equally to this work. Correspondence and requests for materials should be addressed to M.D. (email: marcel.deponte@gmx.de).

Glutaredoxins exert central physiological functions including glutathione-dependent redox catalysis, the biosynthesis of iron–sulfur clusters as well as iron- and redox sensing. In accordance with such a variety of functions, isoforms of this heterogeneous protein family are found in many prokaryotes as well as in the cytosol, nucleus, mitochondria, chloroplasts and/or secretory pathway of eukaryotes[1–8]. Fusion constructs between glutaredoxins and mutated fluorescent proteins furthermore provide valuable genetically encoded sensors for non-invasive redox measurements *in vivo*[9]. A better understanding of the structure–function relationships of glutaredoxins has therefore numerous physiological and even biotechnological implications.

All glutaredoxins share a thioredoxin fold. Subfamilies and isoforms can be categorized based on the sequence similarity, domain architecture, quaternary structure, enzymatic activity, iron–sulfur cluster binding and the number of active site cysteine residues[1–5]. For example, homologues with CxxS- and CxxC-motifs at the active site are classified as monothiol and dithiol glutaredoxins, respectively. Many glutaredoxins catalyse thiol-disulfide exchange reactions with reduced glutathione (GSH) as electron donor. Electron acceptors are glutathionylated disulfide substrates (GSSR)[1–3,10–12] or non-glutathione disulfide substrates such as *Escherichia coli* ribonucleotide reductase (RSSR')[13–15] (Fig. 1a). Presence, activity and properties of glutaredoxins are often analysed in coupled spectrophotometric reductive assays with bis(2-hydroxyethyl)disulfide (HEDS) as a non-glutathione substrate[10–12,15–18] or L-cysteine-glutathione disulfide (GSSCys) as a glutathionylated substrate[10–12,18–21] (Fig. 1a). On the basis of such standard assays, different isoforms are hereinafter referred to as 'enzymatically active or inactive glutaredoxins' for the sake of simplicity (without excluding the possibility that inactive isoforms might actually catalyse other reactions with specialized substrates *in vivo*). Of the eight glutaredoxins in *Saccharomyces cerevisiae*[11,12], ScGrx6 and ScGrx7 are the only monothiol isoforms that are active in standard assays[11,12,18,22–24]. In contrast to dithiol glutaredoxins and ScGrx6/7, other monothiol glutaredoxins from a variety of sources do not possess an enzymatic activity in these assays[25–29]. The underlying structure–function relationships for the different properties and the exact catalytic mechanisms are still elusive[3]. To address the major differences in glutaredoxin activities, we previously suggested two nonexclusive working models for glutaredoxin catalysis termed 'glutathione scaffold' and 'glutathione activator' model. In these models, we take into account the reaction geometry of common thiol-disulfide exchange reactions and distinguish hypothetical protein areas that might either interact with the disulfide substrate (a scaffold site) or the reducing agent (an activator site) (Fig. 1b)[3,12].

Here, we addressed our mechanistic models experimentally using ScGrx7 as a reference enzyme[11,12,18]. ScGrx7 is particularly suited for unbiased kinetic and mechanistic studies because of its high activity, the absence of iron–sulfur clusters in enzyme preparations and the lack of additional cysteine residues[11] (glutaredoxins with two or more cysteine residues have complex kinetics because of alternative redox states and reactions[10,12,14,17,30–33]). First, we selected candidate residues for the potential glutathione activator site and the glutathione-scaffold site. Mutants of the candidate residues were subsequently compared with wild-type ScGrx7 in steady-state kinetic measurements using HEDS and GSSCys as a non-glutathione and glutathione disulfide substrate, respectively. Comparison of the kinetic parameters revealed that ScGrx7 has two distinct glutathione interaction sites. To test a general applicability of this mechanistic model, we confirmed our findings for the non-related enzyme PfGrx from the malaria parasite *Plasmodium falciparum*[18,33]. Furthermore, using redox-sensitive GFP2 (roGFP2)[9,34,35] as a tool for functional analysis of glutaredoxins, we determined that the inactive monothiol glutaredoxin AtGrxS15 from *Arabidopsis thaliana*[36] is able to utilize glutathione disulfide (GSSG) but not GSH as a substrate. The requirement of two distinct glutathione interaction sites for the efficient reduction of GSSR by GSH explains the deviating properties and substrate preferences of glutaredoxin subfamilies as well as thioredoxins with implications for the design and optimization of artificial enzymes and inhibitors.

## Results

**Selection of residues and generation of Grx mutants**. As reviewed recently[3], structural studies on glutaredoxins revealed several residues that are involved in glutathione interactions (Fig. 1c,d). However, a functional assignment of these residues remains difficult because of the absence of structures with transition-state analogues that allow the discrimination between the glutathione moieties from the GSSR substrate or GSH (Fig. 1b). Residue Lys105 of ScGrx7 is conserved in many glutaredoxins[3,11,12,19] ($r_1$ in Fig. 1c,d) and was previously suggested as a candidate for a potential activator site, first, because its positive charge might stabilize or bind GSH as GS⁻, and, second, because of its evolutionary conserved position that is equivalent to the GSH-interacting catalytic residue of glutathione transferases[3]. Furthermore, the residue could contribute to the stabilization of the thiolate of the catalytic cysteine as reported for human Grx1 (ref. 37) and the NrdH-redoxin from *Corynebacterium glutamicum*[38]. In accordance with the glutathione activator hypothesis, only positively charged amino acids replace Lys105 in enzymatically active glutaredoxins, whereas in inactive isoforms and other members of the thioredoxin superfamily the activator candidate is often separated from the catalytic cysteine residue in the primary sequence or is replaced by uncharged residues[3,11,12] (Fig. 1c). To address the role of the activator candidate for glutaredoxin catalysis, we replaced Lys105 of ScGrx7 by site-directed mutagenesis. Arg, Ala, Glu and Tyr were selected as replacements to distinguish the relevance of the charge and size of the side chain and to mimic tyrosine-dependent glutathione transferases[3,18].

A candidate for a potential glutathione-scaffold residue was identified using the following selection criteria: The residue should be rather conserved in enzymatically active in contrast to inactive glutaredoxins and have a charged side chain to ensure a strong ionic glutathione interaction (Fig. 1c,d). Furthermore, we took into account that the glutathione moiety that originates from the GSSR disulfide substrate might be sandwiched between the protein and the glutathione moiety of GSH approaching the active site (Fig. 1b,d). We therefore selected Glu170 ($r_6$ in Fig. 1c,d) instead of one of the residues sticking out on the protein surface to avoid combined interactions with GSSR and GSH (Fig. 1d). To elucidate the relevance of charge and size of the side chain of the scaffold candidate, we replaced Glu170 of ScGrx7 with Asp, Ala or Lys. Recombinant Lys105 and Glu170 mutants were purified from *E. coli* with very similar yields and purities (Supplementary Fig. 1). Freshly purified proteins were subsequently analysed in steady-state kinetic measurements using GSSCys and HEDS as alternative disulfide substrates.

**Lys105 is a GSH and enzyme activator in the GSSCys assay**. In a first set of experiments, we analysed the effects of the Lys105 replacements on the steady-state kinetics at variable GSSCys and GSH concentrations. Wild-type ScGrx7 was studied

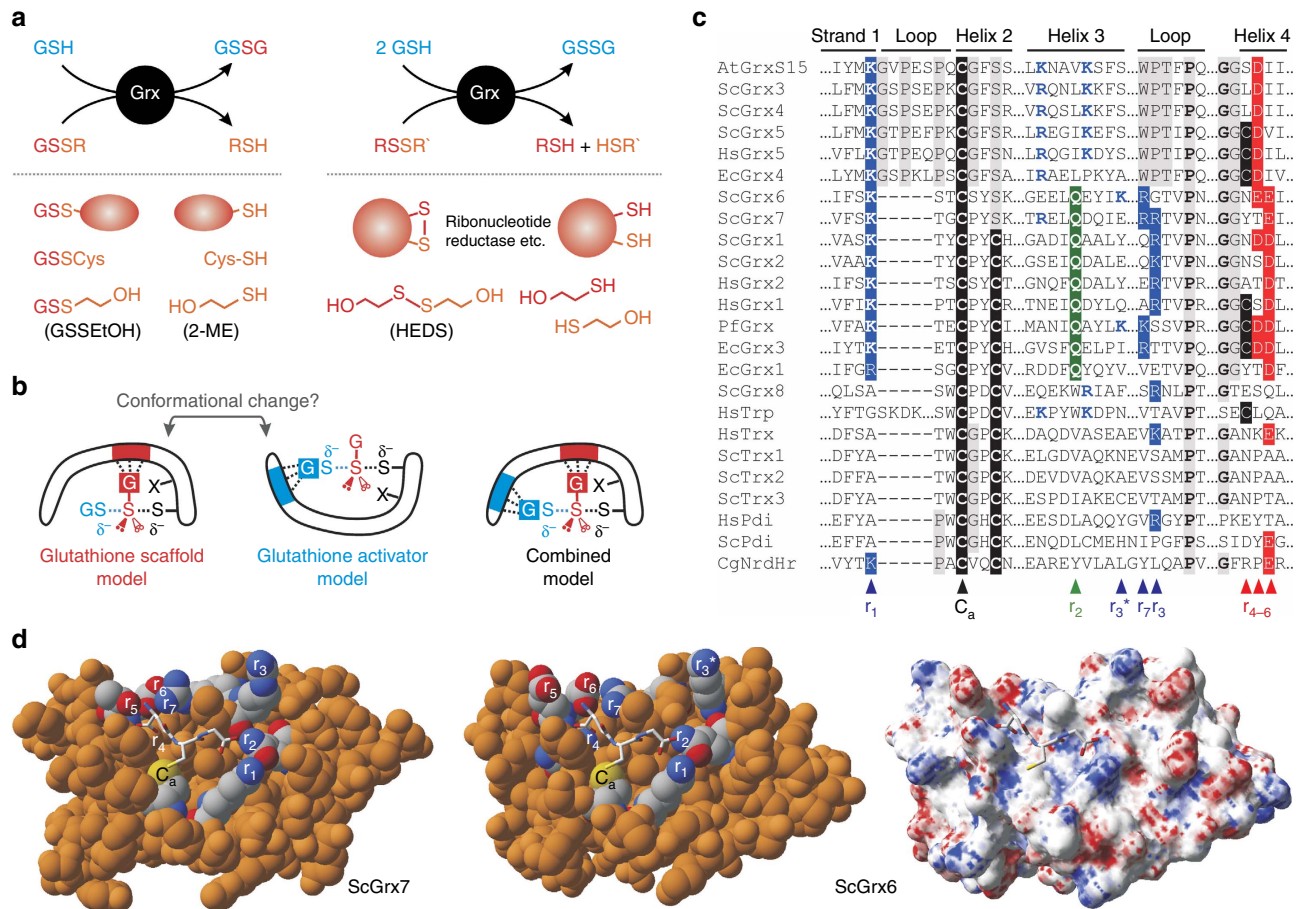

**Figure 1 | Structure–function relationships of glutaredoxins. (a)** Enzymatically active glutaredoxins (Grx) use GSH as an electron donor for the reduction of high- and low-molecular weight glutathione disulfide substrates (left side) or non-glutathione disulfide substrates (right side). **(b)** Two nonexclusive models for glutaredoxin catalysis are based on the reaction geometry of the transition states of thiol-disulfide exchange reactions. The models distinguish protein–substrate interactions with the glutathione moieties of GSSR in red and GSH in blue[3,12]. Only the transition state yielding GSSG is shown for the sake of simplicity. **(c)** Sequence alignment of glutaredoxin isoforms and comparison with other proteins of the thioredoxin superfamily from *A. thaliana* (At), *S. cerevisiae* (Sc), *Homo sapiens* (Hs), *P. falciparum* (Pf), *E. coli* (Ec) and *C. glutamicum* (Cg). The manual alignment is based on structural overlays and comparisons of PDB entries 2WCI, 3L4N, 3D4M, 3D5J, 2M80, 2WUL, 2WOU, 1MEK, 1B4Q and 4FIW. **(d)** Comparison between models of ScGrx7 and ScGrx6 with potential glutathione-interacting residues highlighted[11]. The structure of ScGrx6 was confirmed by Luo *et al.* (PDB entry 3L4N)[24]. The electrostatic potential was computed and mapped to the protein surface using the Poisson-Boltzmann method of Swiss-PDB Viewer at 0.1 M solvent ionic strength (colouring: red − 4, blue + 4). Orientation and numbering of potential glutathione-interacting residues $r_{1-7}$ in **c** and **d** are based on previous presentations[3,11] to facilitate a comparison.

in parallel and served as a control. Regression and pattern analyses revealed ping-pong kinetics for all mutants (Supplementary Fig. 2), indicating that the general mechanism with a separate oxidative and reductive half-reaction was not altered by the mutations. Replacement of Lys105 by uncharged residues in K105A/Y resulted in a 65–97% decrease of the $k_{cat}^{app}$ values for GSSCys and GSH (Fig. 2a), suggesting that the residue affects both the oxidative half-reaction with GSSCys and the reductive half-reaction with GSH. The charge inversion in K105E further enhanced the effects. Moreover, replacements of Lys105 resulted in a reciprocal increase of the $K_m^{app}$(GSSCys) values and decrease of the $K_m^{app}$(GSH) values (Fig. 2b) as well as a decrease of the apparent catalytic efficiencies (Fig. 2c). Wild-type enzyme and K105X mutants tended to have infinite true $k_{cat}$ and $K_m$ values for extrapolated infinite substrate concentrations, suggesting that the enzymes are neither saturated by GSSCys nor GSH (Supplementary Fig. 3; Supplementary Table 1). The velocity of such a ping-pong reaction with infinite true $k_{cat}$ and $K_m$ values can be described by the equation $[E]/V = \Phi_1/[\text{GSSCys}] + \Phi_2/[\text{GSH}]$, wherein the Dalziel coefficients $\Phi_1$ and

$\Phi_2$ are the slopes of secondary plots for $k_{cat}^{app}$(GSH) and $k_{cat}^{app}$(GSSCys), respectively[39] (Supplementary Table 1). The advantage of this data evaluation is that the reciprocal coefficients $1/\Phi_1$ and $1/\Phi_2$ can be interpreted as apparent rate constants $k_{ox}^{app}$ and $k_{red}^{app}$ for the oxidative and reductive half-reaction, respectively. Both $1/\Phi$ values were similar to the corresponding apparent catalytic efficiencies and decreased for K105X mutants by up to one or two orders of magnitude (Fig. 2c,d). The effects of the replacement of Lys105 on the kinetic parameters of K105A/Y/E are summarized in Fig. 2e.

Our data interpretation, based on a general ping-pong mechanism, is summarized in Fig. 2f. Lys105 has a dual function for the oxidative and reductive half-reaction of ScGrx7 resulting in a decrease of all $k_{cat}^{app}$, $k_{cat}^{app}/K_m^{app}$ and $1/\Phi$ values for K105A/Y/E (Fig. 2e). The reductive half-reaction with GSH was probably rate-limiting for wild-type enzyme ($k_{ox}^{app} \approx 1/\Phi_1 > 1/\Phi_2 \approx k_{red}^{app}$), whereas for all K105X mutants the oxidative half-reaction with GSSCys became slower ($1/\Phi_1 < 1/\Phi_2$) (Fig. 2d). All $k_{cat}^{app}/K_m^{app}$ and $1/\Phi$ values were $< 10^6 \text{ M}^{-1}\text{s}^{-1}$ and too slow for a diffusion-controlled reaction

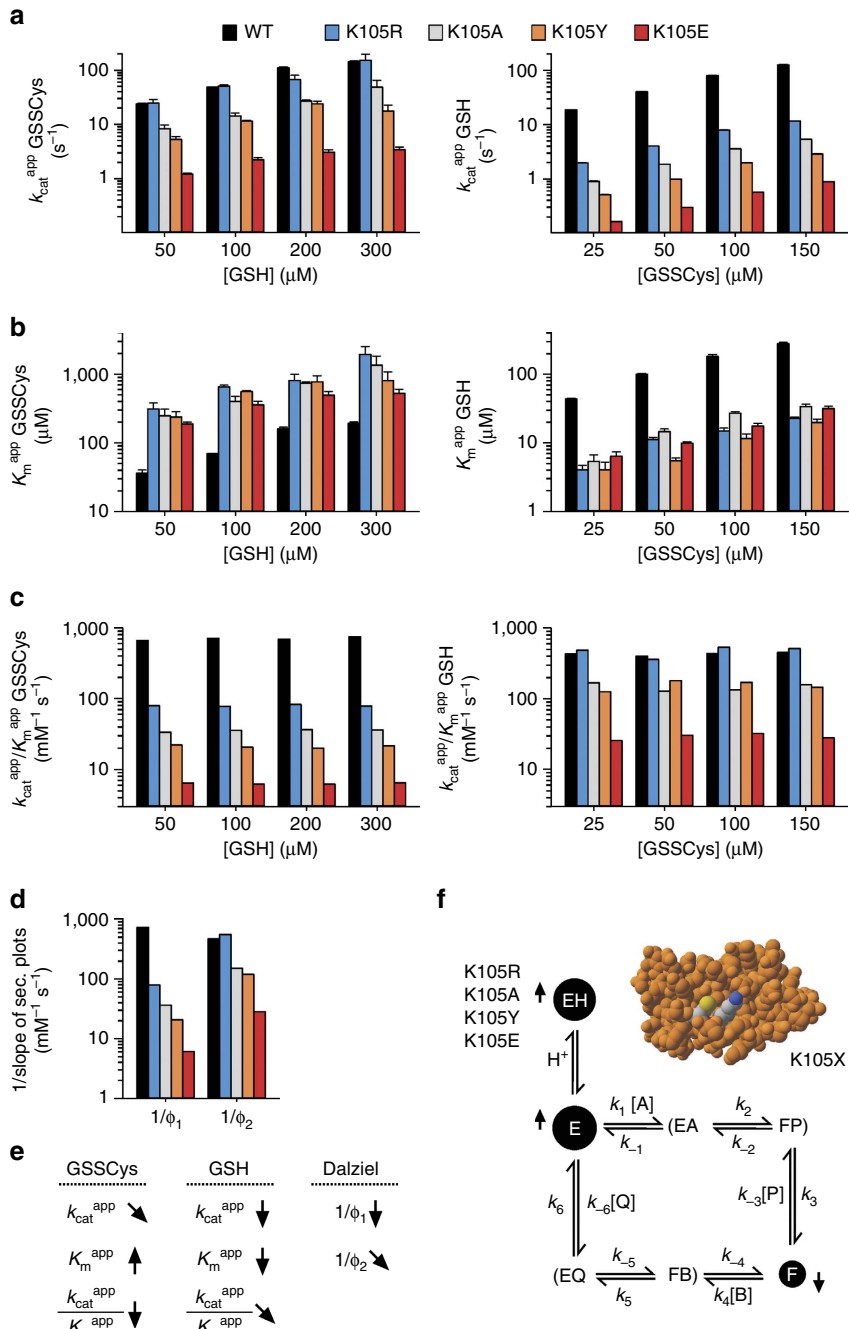

**Figure 2 | Lys105 is crucial for the oxidative and reductive half-reaction of the GSSCys assay.** (**a,b**) Selected $k_{cat}^{app}$ and $K_m^{app}$ values of ScGrx7 wild-type enzyme and K105X mutants for GSSCys and GSH. (**c**) Calculated catalytic efficiencies from **a** and **b**. (**d**) Reciprocal Dalziel coefficients, which probably reflect the rate constants of the oxidative and reductive half-reaction with GSSCys and GSH, respectively. (**e**) Summary of the altered kinetic parameters for K105A/Y/E. (**f**) Data interpretation based on a general ping-pong mechanism. Altered concentrations of the active free enzyme 'E' (thiolate form), glutathionylated enzyme 'F' and inactive protonated enzyme 'EH' upon mutation of Lys105 are highlighted. GSSCys and GSH correspond to substrates A and B, respectively. Cysteine and GSSG correspond to products P and Q, respectively. Original plots and kinetic parameters for **a–c** are shown in Supplementary Fig. 2 and Supplementary Table 2 and are the mean ± s.d. from at least three independent replicates. Statistical analyses and *P*-values for the $k_{cat}^{app}$ and $K_m^{app}$ values from **a** and **b** are listed in Supplementary Table 11. Reciprocal Dalziel coefficients for **d** were obtained from Supplementary Fig. 3 and are listed together with the true $k_{cat}$ values in Supplementary Table 1.

around $10^8$–$10^9$ M$^{-1}$s$^{-1}$. The infinite true $k_{cat}$ and $K_m$ values therefore suggest a rate-limiting productive interaction between substrate and enzyme followed by a rapid substrate turnover without accumulation of an enzyme–substrate complex. Hence, the reciprocal changes of $K_m^{app}{}_{(GSSCys)}$ and $K_m^{app}{}_{(GSH)}$ in Fig. 2b do not reflect true substrate affinities but rather indicate an

altered steady-state equilibrium between the thiolate form 'E' and the glutathionylated form 'F' of the enzyme (Fig. 2f). In other words, $K_m^{app}{}_{(GSSCys)}$ and $K_m^{app}{}_{(GSH)}$ are not solely defined by the ratios $k_{-1}/k_1$ and $k_{-4}/k_4$ but are also affected by other rate constants in Fig. 2f. For example, the tenfold lower $K_m^{app}{}_{(GSH)}$ of K105E did not result from a higher affinity for GS$^-$

(this mutation should have actually decreased the affinity and $k_4/k_{-4}$ ratio for the negatively charged substrate) but from lower steady-state concentrations of the glutathionylated enzyme 'F', yielding apparent substrate saturations at lower GSH concentrations (Fig. 2b). Likewise, all K105X mutants had increased $K_m^{app}$(GSSCys) values, suggesting an apparent substrate saturation at higher GSSCys concentrations because of an increased steady-state concentration of the thiolate form 'E'. The drastic charge-dependent decrease of the $1/\Phi_2$ values of K105X mutants indicates that Lys105 is crucial for an efficient interaction with GSH during the reductive half-reaction in accordance with our glutathione activator model. The drastic deceleration of the oxidative half-reaction of K105X mutants furthermore suggests that the residue also stabilizes the enzyme thiolate form 'E'. For example, mutation of Lys105 might have increased the steady-state concentration of an inactive protonated enzyme form 'EH', resulting in less frequent productive encounters between the enzyme and GSSCys (Fig. 2f). A distinguished dual role of Lys105 as a glutaredoxin and glutathione activator is also supported by anomalies that were observed for K105R: In contrast to the other K105X mutants, values for $k_{cat}^{app}/K_m^{app}$(GSH) and $1/\Phi_2$ were highly similar between K105R and the wild-type enzyme. Replacement of lysine by arginine therefore preserved the GSH activator function but not the glutaredoxin activator function (as reflected by the decreased $1/\Phi_1$ values). In summary, mutation of Lys105 affected the oxidative and reductive half-reaction in different ways and resulted in decreased steady-state concentrations of the glutathionylated enzyme and much slower turnover of both substrates. Thus, Lys105 exerts a charge- and shape-dependent dual function as a glutaredoxin and glutathione activator.

**Lys105 is a GSH and enzyme activator in the HEDS assay**. We subsequently compared the steady-state kinetics of K105X mutants and wild-type ScGrx7 at variable HEDS and GSH concentrations. Replacement effects are again summarized in the first paragraph followed by our data interpretation in the second paragraph: Regression and pattern analyses revealed sequential kinetics for all mutants with an apparent common intersection point at the $x$ axis (Supplementary Fig. 4). Replacement of Lys105 by uncharged residues resulted in a 92–98% decrease of the $k_{cat}^{app}$ values for HEDS and GSH (Fig. 3a). Charge inversion by glutamate substitution further enhanced the effects. In contrast, all K105X mutants appeared to have similar $K_m^{app}$ values for HEDS and GSH, so that the apparent catalytic efficiencies decreased according to the altered $k_{cat}^{app}$ values (Fig. 3b,c). Secondary plots suggested a type- and charge-dependent decrease of the true $k_{cat}$ values for the K105X mutants. The intercepts were close to the origin (Supplementary Fig. 5; Supplementary Table 1). The true $K_m$ values seemed to be rather constant regardless of the mutation. Reciprocal Dalziel coefficients $1/\Phi_1$ and $1/\Phi_2$ from the slopes of secondary plots decreased for both substrates from $\geq 10^5 \, M^{-1} s^{-1}$ by up to two orders of magnitude (Fig. 3d). The altered kinetic parameters of K105X mutants are schematically summarized in Fig. 3e.

Our data interpretation is visualized in Fig. 3f. The exact enzymatic mechanism of the HEDS assay is still unknown. Previous analyses indicated that the non-enzymatic formation of the mixed disulfide between GSH and 2-mercaptoethanol (GSSEtOH) is too slow to explain the high reaction velocities[18]. We therefore suggested an enzymatic turnover of HEDS with a mixed disulfide between glutaredoxin and 2-mercaptoethanol as reaction intermediate (enzyme 'E•' in Fig. 3f) as originally proposed by Mieyal et al.[17] Regardless of the true disulfide substrate, the type- and charge-dependent decrease of all $1/\Phi$,

$k_{cat}^{app}/K_m^{app}$ and $k_{cat}^{app}$ values for the K105X mutants are in agreement with a dual function of Lys105 as a glutaredoxin and glutathione activator. Lys105 replacement increased the concentration of inactive enzyme 'EH' and reduced the reactivity of 'E' with HEDS and of 'F' with GSH. Furthermore, if 'E•' is catalytically relevant and reacts with GSH in agreement with the glutathione activator model[18], GSSEtOH would be formed as a product and could subsequently react as a substrate ('B•'), yielding the glutathionylated enzyme species 'F' (Fig. 3f). (A direct formation of 'F' from 'E•' without formation of GSSEtOH is unlikely because of the reaction geometry and accessibility of the enzyme sulfur atom in 'E•' as outlined below). The first part of the HEDS assay would then comprise an additional sequence of thiol-disulfide exchanges and the $1/\Phi_2$ values could reflect the GSH-dependent reduction of either 'E•' or 'F'. The reduction of 'E•' appears to be more plausible because the $1/\Phi_2$ values for the HEDS assay were all smaller than for the GSSCys assay despite identical reactions between 'F' and GSH in both assays (Figs 2d,f and 3d,f, Supplementary Table 1). A model in which HEDS and GSH both interact with the enzyme during a rate-limiting reaction sequence that yields 'F' could also explain the absence of ping-pong patterns and that the $K_m^{app}$ values were not reciprocally altered by the Lys105 mutations. In summary, we propose a catalytic model for the HEDS assay with two different covalent enzyme modifications, GSSEtOH as a product and substrate, and Lys105 as a glutaredoxin and glutathione activator.

**Glu170 affects the interaction with GSSCys**. In a second independent set of experiments, we analysed the effects of the Glu170 replacements on the steady-state kinetics at variable GSSCys and GSH concentrations. Wild-type ScGrx7 was studied again in parallel and served as a control for systematic variations. Replacement effects are summarized in the first paragraph followed by our data interpretation in the second paragraph: Regression and pattern analyses revealed ping-pong kinetics for all mutants (Supplementary Fig. 6). Effects for the E170X mutants on the determined kinetic parameters were in general far less pronounced than for the K105X mutants. Replacement of Glu170 by alanine or lysine resulted in a 40–50% decrease of $k_{cat}^{app}$(GSH) in contrast to $k_{cat}^{app}$(GSSCys) (Fig. 4a). $K_m^{app}$ values for GSSCys and GSH were reciprocally doubled and halved, respectively (Fig. 4b). Catalytic efficiencies $k_{cat}^{app}/K_m^{app}$(GSSCys) and $1/\Phi_1$ values of E170A/K were both decreased by 40–45% in contrast to the $k_{cat}^{app}/K_m^{app}$(GSH) and $1/\Phi_2$ values (Fig. 4c,d). All E170X mutants tended to have infinite true $k_{cat}$ and $K_m$ values for extrapolated infinite substrate concentrations (Supplementary Fig. 7; Supplementary Table 1). The altered kinetic parameters of E170A/K are schematically summarized in Fig. 4e.

Our data interpretation is visualized in Fig. 4f. Glu170 is far away from the active site cysteine residue and had probably no direct effect on $GS^-$ stabilization and enzyme thiolate formation. The decrease of the $k_{cat}^{app}/K_m^{app}$(GSSCys) and $1/\Phi_1$ values for E170A/K implies a decelerated oxidative half-reaction, which resulted in increased steady-state concentrations of the thiolate form 'E' and decreased concentrations of the glutathionylated form 'F' (Fig. 4f). Hence, more GSSCys and less GSH were required for apparent substrate saturation under steady-state conditions as reflected by the reciprocal $K_m^{app}$ values (Fig. 4c). The accumulation of 'E' for E170A/K was probably caused by a less efficient interaction between 'E' and the glutathionyl moiety of GSSCys in accordance with the glutathione-scaffold model. All values for $k_{cat}^{app}/K_m^{app}$(GSSCys) and $1/\Phi_1$ were greater than for $k_{cat}^{app}/K_m^{app}$(GSH) and $1/\Phi_2$ regardless of the replacement (Fig. 4c,d). This suggests that the reductive half-reaction remained rate-limiting for all E170X mutants

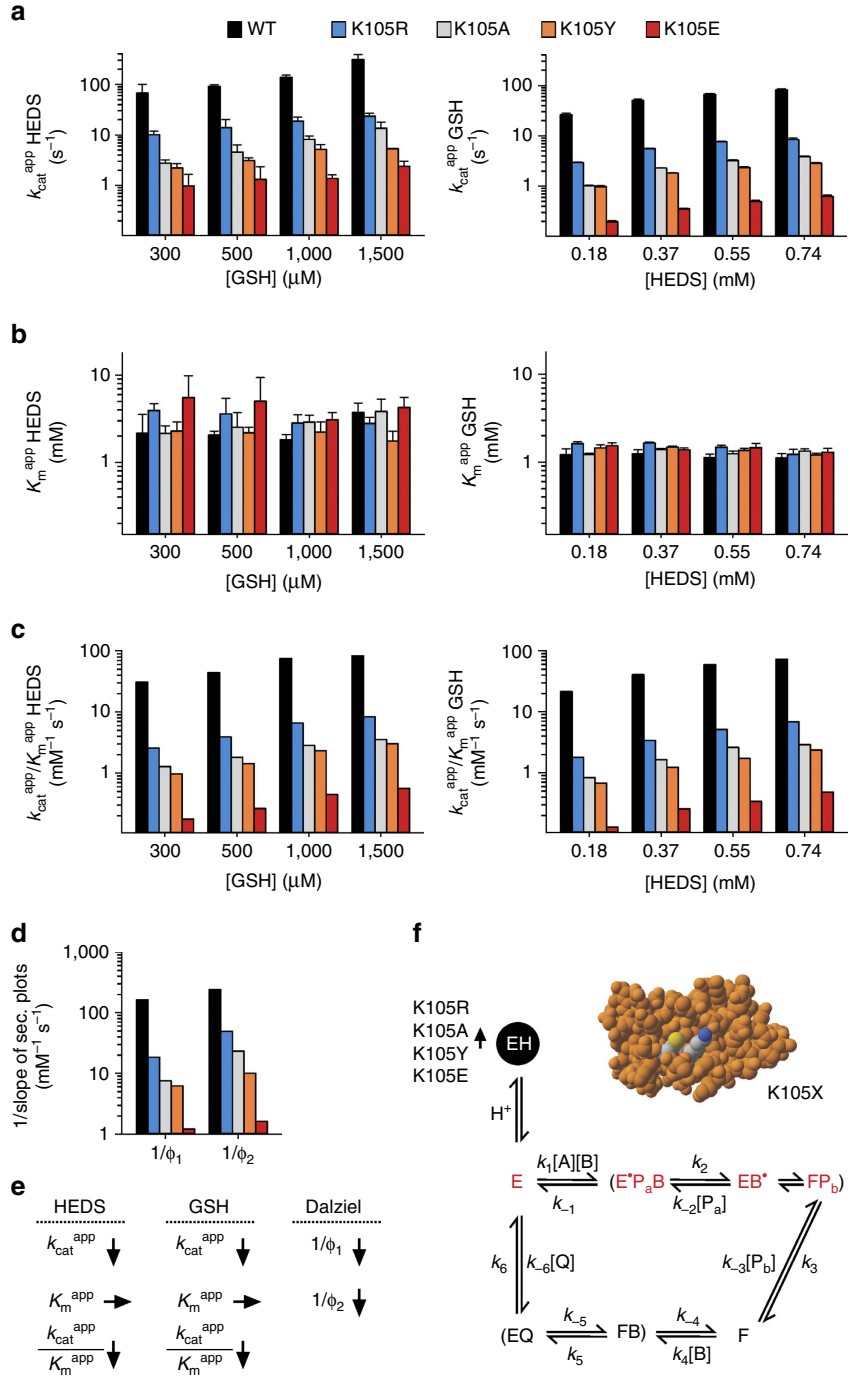

**Figure 3 | Replacement of Lys105 slows down the turnover of both substrates in the HEDS assay.** (**a,b**) Selected $k_{cat}^{app}$ and $K_m^{app}$ values of ScGrx7 wild-type enzyme and K105X mutants for HEDS and GSH. (**c**) Calculated catalytic efficiencies from **a** and **b**. (**d**) Reciprocal Dalziel coefficients obtained from Supplementary Fig. 5. (**e**) Summary of the altered kinetic parameters. (**f**) Data interpretation based on a proposed catalytic mechanism with HEDS and GSH as true substrates[18]. The altered concentration of the inactive protonated enzyme 'EH' upon mutation of Lys105 is highlighted. Kinetically relevant enzyme species are labelled in red and comprise the active free enzyme 'E' (thiolate form), a mixed disulfide between the enzyme and 2-mercaptoethanol 'E•' and glutathionylated enzyme 'F'. HEDS and GSH correspond to substrates A and B, respectively. GSSEtOH is the intermediate product and substrate 'B•'. Two molecules of 2-ME and GSSG correspond to products $P_{a,b}$ and Q, respectively. Original plots and kinetic parameters for **a–c** are shown in Supplementary Fig. 4 and Supplementary Table 3 and are the mean ± s.d. from at least three independent replicates. Statistical analyses and P-values for the $k_{cat}^{app}$ and $K_m^{app}$ values from **a** and **b** are listed in Supplementary Table 11. Reciprocal Dalziel coefficients for **d** and true $k_{cat}$ values are listed in Supplementary Table 1.

$(k_{ox}^{app} \approx 1/\Phi_1 > 1/\Phi_2 \approx k_{red}^{app})$ and explains the rather minor effects of the Glu170 replacements as compared with the K105X mutants. In summary, steady-state kinetics for E170A/K revealed an impaired GSSCys interaction in accordance with the glutathione-scaffold model. The impaired interaction decelerated the oxidative half-reaction and decreased the steady-state concentration of available glutathionylated enzyme for the rate-limiting turnover of GSH.

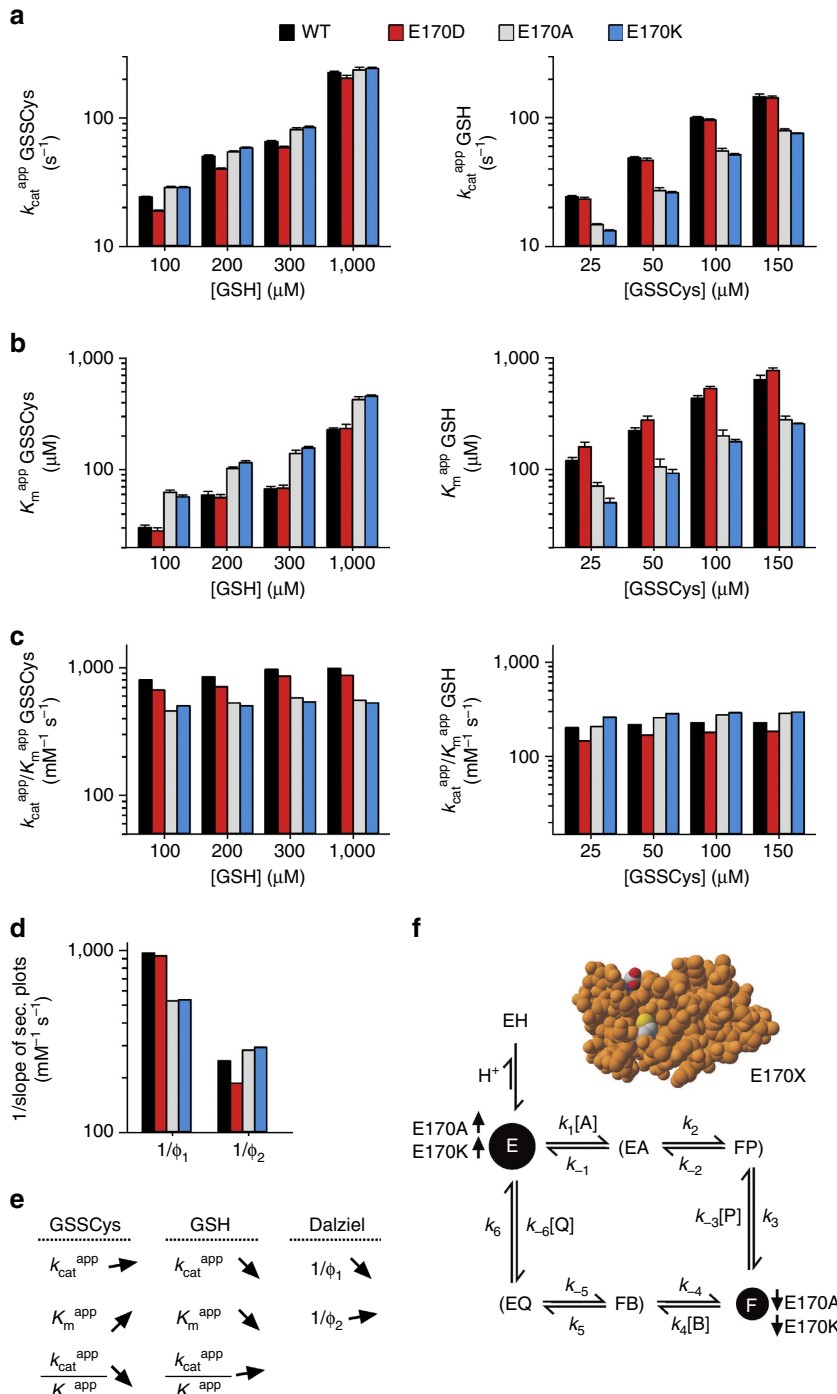

**Figure 4 | Replacement of Glu170 decelerates the oxidative half-reaction of the GSSCys assay.** (**a,b**) Selected $k_{cat}^{app}$ and $K_{m}^{app}$ values of ScGrx7 wild-type enzyme and E170X mutants for GSSCys and GSH. (**c**) Calculated catalytic efficiencies from **a** and **b**. (**d**) Reciprocal Dalziel coefficients, which probably reflect the rate constants of the oxidative and reductive half-reaction with GSSCys and GSH, respectively. (**e**) Summary of the altered kinetic parameters for E170A/K. (**f**) Data interpretation based on a general ping-pong mechanism. Altered concentrations of the active free enzyme 'E' (thiolate form) and glutathionylated enzyme 'F' upon mutation of Glu170 are highlighted. GSSCys and GSH correspond to substrates A and B, respectively. Cysteine and GSSG correspond to products P and Q, respectively. Original plots and kinetic parameters for **a–c** are shown in Supplementary Fig. 6 and Supplementary Table 4 and are the mean ± s.d. from at least three independent replicates. Statistical analyses and P-values for the $k_{cat}^{app}$ and $K_{m}^{app}$ values from **a** and **b** are listed in Supplementary Table 11. Reciprocal Dalziel coefficients for **d** were obtained from Supplementary Fig. 7 and are listed together with the true $k_{cat}$ values in Supplementary Table 1.

**Glu170 interacts with HEDS-derived GSSEtOH.** Next, we compared the steady-state kinetics of E170X mutants and wild-type ScGrx7 at variable HEDS and GSH concentrations. Since HEDS contains no glutathione moiety, alterations at the

scaffold site could provide further insights regarding GSSEtOH as a potential reaction intermediate. Replacement effects are summarized in the first paragraph followed by our data interpretation in the second paragraph: Regression and pattern

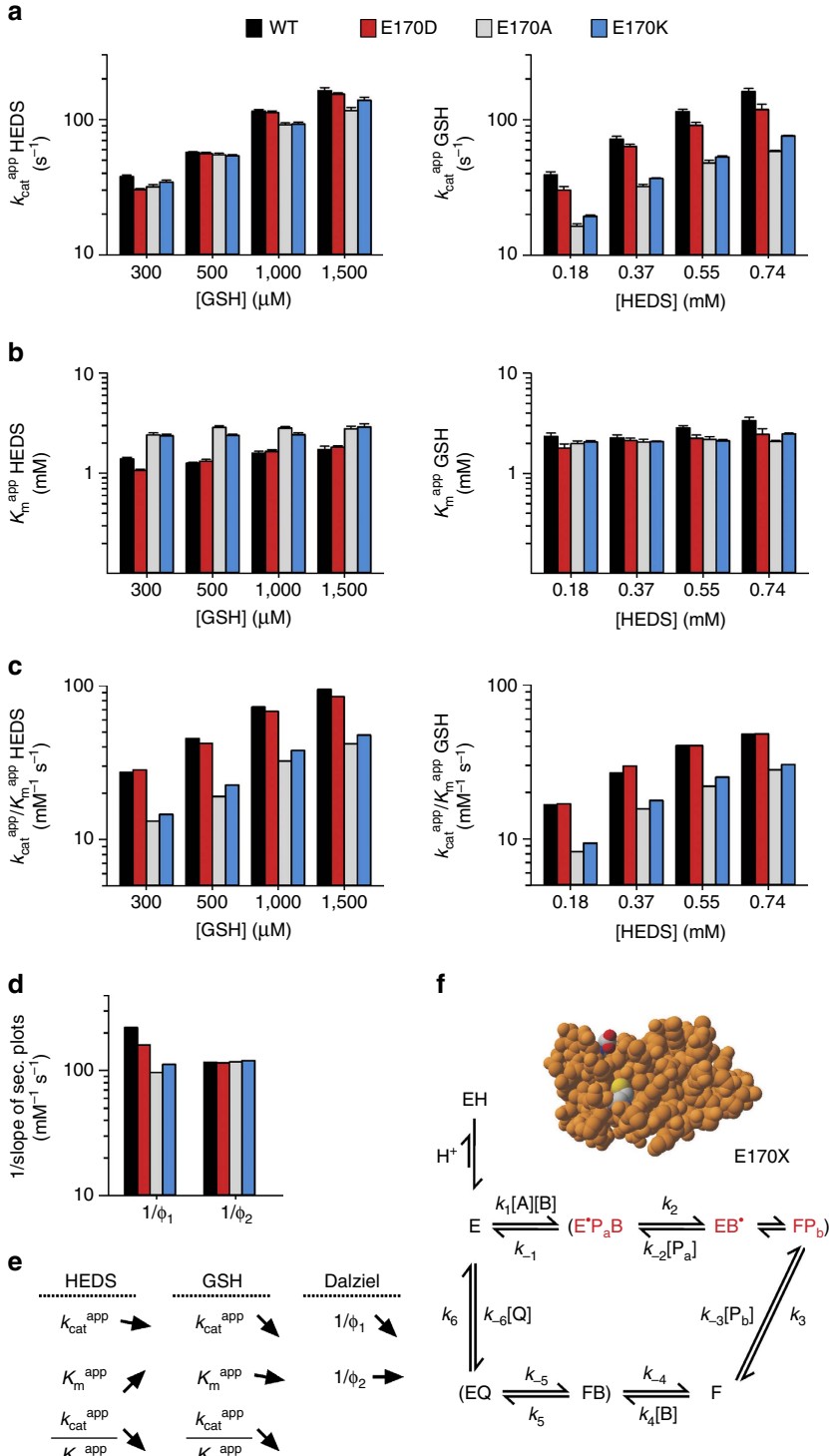

**Figure 5 | Altered kinetics of E170X mutants support a glutathionylated disulfide substrate intermediate.** (**a,b**) Selected $k_{cat}^{app}$ and $K_m^{app}$ values of ScGrx7 wild-type enzyme and E170X mutants for HEDS and GSH. (**c**) Calculated catalytic efficiencies from **a** and **b**. (**d**) Reciprocal Dalziel coefficients obtained from Supplementary Fig. 9. (**e**) Summary of the altered kinetic parameters. (**f**) Data interpretation based on a proposed catalytic mechanism with HEDS and GSH as true substrates[18]. Kinetically relevant enzyme species are labelled in red and comprise the mixed disulfide between the enzyme and 2-mercaptoethanol 'E•', an intermediate thiolate form 'E' and glutathionylated enzyme 'F'. HEDS and GSH correspond to substrates A and B, respectively. GSSEtOH is the intermediate product and substrate 'B•'. Two molecules of 2-ME and GSSG correspond to products $P_{a,b}$ and Q, respectively. Original plots and kinetic parameters for **a**–**c** are shown in Supplementary Fig. 8 and Supplementary Table 5 and are the mean ± s.d. from at least three independent replicates. Statistical analyses and *P*-values for the $k_{cat}^{app}$ and $K_m^{app}$ values from **a** and **b** are listed in Supplementary Table 11. Reciprocal Dalziel coefficients for **d** and true $k_{cat}$ values are listed in Supplementary Table 1.

analyses revealed sequential kinetics for all mutants with an apparent common intersection point at the $x$ axis (Supplementary Fig. 8). Replacement of Glu170 in E170A/K resulted in a 50–60% decrease of the $k_{cat}^{app}$ values for GSH but had a rather minor effect on the $k_{cat}^{app}$(HEDS) values (Fig. 5a). Moderate changes of the $K_m^{app}$ values in combination with the altered $k_{cat}^{app}$ values resulted in decreased apparent catalytic efficiencies of E170A/K for both substrates as compared with the wild-type enzyme and E170D (Fig. 5b,c). Secondary plots revealed a type- and charge-dependent decrease of the true $k_{cat}$ value for the E170X mutants. The intercepts were very close to the origin (Supplementary Fig. 9; Supplementary Table 1). Mutation of Glu170 decreased the reciprocal Dalziel coefficient $1/\Phi_1$ by up to 56% whereas $1/\Phi_2$ remained constant (Fig. 5d). In contrast to the GSSCys assay, the $k_{cat}^{app}/K_m^{app}$ values highly depended on the substrate concentration and differed from the $1/\Phi$ values by up to one order of magnitude. Effects of the replacement of Glu170 are summarized in Fig. 5e.

Our data interpretation is visualized in Fig. 5f. Glu170 is too far away from the active site to directly affect the interaction between HEDS and the cysteine thiolate. The decrease of the $1/\Phi_1$ values (Fig. 5d) therefore suggests a less efficient interaction between E170X mutants and HEDS-derived GSSEtOH in accordance with the scaffold model and the hypothesis that GSSEtOH formation is catalysed by ScGrx7. Furthermore, constant $1/\Phi_2$ values for all E170X mutants (in contrast to K105X mutants) suggest that the mutation of Glu170 had no effect on a rate-limiting reduction by GSH (Fig. 5d). As outlined for the K105X mutants, this might be the reduction of enzyme form 'E•' (Fig. 5f), because the rate constants for the reduction of 'F' in the GSSCys assay were always larger than the $1/\Phi_2$ values from the HEDS assay (Supplementary Table 1). The irrelevance of the reduction of 'F' for the HEDS assay kinetics could again explain the sequential patterns and non-reciprocal $K_m^{app}$ values in contrast to the GSSCys assays. In summary, the kinetic data for the E170X mutants support a glutaredoxin-catalysed formation of GSSEtOH as well as an exclusive interaction between Glu170 and the glutathione moiety of GSSR in accordance with the glutathione-scaffold model.

**ScGrx7 is poorly susceptible to competitive inhibitors.** To test the catalytic mechanism in more detail, we performed inhibition studies with S-methylglutathione or L-γ-glutamyl-L-α-aminobu-tyrylglycine (ophthalmic acid). Both inhibitors were added to HEDS or GSSEtOH assays[18]. No significant inhibitory effects on ScGrx7 catalysis were observed at inhibitor concentrations of up to 1.5 mM. Only when we tested up to 10 mM ophthalmic acid, the reaction velocity was reduced by 10–20% (Supplementary Fig. 10). Thus, non-reactive substrate analogues are very weak competitive inhibitors of ScGrx7.

**Kinetics of PfGrx mutants confirm a conserved mechanism.** In a third set of experiments, we tested whether our findings for ScGrx7 catalysis can be generalized. We therefore repeated all experiments for the non-related enzyme PfGrx from the malaria parasite P. falciparum[18,33]. First, we replaced the homologous residues Lys26 and Asp90 ($r_1$ and $r_6$ in Fig. 1c) of PfGrx$^{C32S/C88S}$ by alanine. We chose the monothiol mutant PfGrx$^{C32S/C88S}$ for our studies to avoid side reactions that complicate the steady-state kinetics[33]. The recombinant mutants K26A and D90A were subsequently purified (Supplementary Fig. 1c) and analysed in GSSCys and HEDS assays with PfGrx$^{C32S/C88S}$ as a control. Detected effects regarding the kinetic parameters of K26A and D90A were highly similar to the ScGrx7 mutants K105A and E170A, respectively (Supplementary Figs 11–16; Supplementary Tables 6–8). For example, the $1/\Phi_1$ and $k_{cat}^{app}/K_m^{app}$(GSSCys)

values of K26A and K105A were both decreased by 95%, whereas the $1/\Phi_2$ and $k_{cat}^{app}/K_m^{app}$(GSH) values of K26A and K105A in the GSSCys assay were both decreased by ∼65%. Replacement of Asp90 had a rather moderate but significant effect for PfGrx catalysis as expected. For example, the $1/\Phi_1$ and $k_{cat}^{app}/K_m^{app}$(GSSCys) values of D90A decreased by 20–30% in accordance with the glutathione-scaffold model as outlined for ScGrx7. In summary, we confirmed distinct roles of residues $r_1$ and $r_6$ for glutaredoxin catalysis using the non-related enzyme PfGrx. A comparison between the kinetic parameters of mutated ScGrx7 and PfGrx reveals extreme similarities and supports a general applicability of our mechanistic model.

**AtGrxS15 can oxidize but not reduce roGFP2.** To address which half-reaction is inactive for iron–sulfur cluster-binding monothiol glutaredoxins, we analysed A. thaliana GrxS15, which has a CGFS-motif and only one cysteine residue in total (Fig. 1c). The protein was shown to be inactive in the HEDS assay but to react with roGFP2 (ref. 36). Here we used the latter property to monitor the oxidative and reductive half-reaction. Reduced roGFP2 was oxidized much faster by GSSG in the presence of AtGrxS15 as compared with a negative control (Supplementary Fig. 17a). Although AtGrxS15 catalysis was less efficient than for the dithiol glutaredoxin AtGrxC1, the oxidation of roGFP2 clearly depended on the concentration of AtGrxS15. In contrast to the oxidation of reduced roGFP2, AtGrxS15 did not catalyse the reduction of oxidized roGFP2 in the presence of GSH (Supplementary Fig. 17b). A plausible interpretation of the results is that AtGrxS15 was able to react with GSSG and that glutathionylated AtGrxS15 subsequently transferred its glutathione moiety to reduced roGFP2. Thus, the protein appears to have a partially functional glutathione-scaffold site. The fact that AtGrxS15 could not reduce oxidized roGFP2 with the help of GSH might point to an altered or blocked glutathione activator site.

**Role of residue Tyr110 and future active site mapping.** Is it possible to further map the different glutathione interaction sites of ScGrx7 using steady-state kinetics? To address this question, we mutated Tyr110 in the CPYS-motif of ScGrx7 as a candidate residue that might contribute to the glutathione activator site (see Discussion for details) and performed a preliminary study with wild-type ScGrx7 as a control. Replacement of Tyr110 in recombinant Y110A decreased both $k_{cat}^{app}$ and $1/\Phi$ values for the disulfide substrate and GSH in the GSSCys and HEDS assay (Supplementary Figs 18 and 19). This effect is highly similar to the ScGrx7 mutants K105A and K105Y (Figs 2 and 3). However, in contrast to K105A/Y, the $K_m^{app}$ values of Y110A in the GSSCys assay were altered to a much lesser extent and the reductive half-reaction with GSH remained rate-limiting ($k_{cat}^{app}/K_m^{app}$(GSSCys) > $k_{cat}^{app}/K_m^{app}$(GSH) and $1/\Phi_1 > 1/\Phi_2$). Our preliminary interpretation is that replacement of Tyr110 indeed affects the interaction with GSH but, as compared to K105A/Y, does not drastically alter the ratio between the steady-state concentrations of the deprotonated and glutathionylated enzyme species 'E' and 'F', respectively. Thus, residues Tyr110 and Lys105 might share a function in accordance with the glutathione activator model but might also differ, for example, regarding the protonation state of the active site cysteine residue. Further mutants are obviously necessary to decipher the exact role of Tyr110 and to precisely map the glutathione interaction sites in future studies.

**Discussion**

Which glutaredoxin structure–function relationships determine their substrate preferences? The preference of mammalian and

bacterial glutaredoxins for GSSR highly depends on the γ-glutamyl moiety of glutathione[40–42]. Furthermore, human Grx1 reacts twenty times faster with GSH than with L-cysteinylglycine or L-cysteine, revealing that the preference for GSH also depends on its γ-glutamyl moiety[19]. Taking into account the reaction geometry of thiol-disulfide exchange reactions, we raised the question which enzyme area interacts with which substrate[3,12] (Fig. 1b). Here we showed that ScGrx7 has two different glutathione interaction sites, a glutathione-scaffold site, which includes Glu170 and interacts with GSSR, and a glutathione activator site, which includes Lys105 and interacts with GSH. The results were confirmed for the non-related enzyme PfGrx with its homologous residues Lys26 and Asp90. Thus, structures of glutathionylated glutaredoxins[31,43–45] most likely resemble enzyme species 'F', whereas similar structures of complexes between reduced glutaredoxin and GSH with longer sulfur–sulfur distances[24,46] do not represent the interaction with GSH (but probably also resemble species 'F').

On the basis of existing glutaredoxin structures, the previously published model of ScGrx7 (ref. 11) and our kinetic data, we can now assign the glutathione-scaffold site for ScGrx7. The site includes Glu170 as well as candidate residues Gln143 ($r_2$) in helix 3, Arg152 ($r_7$) and Thr154-Pro156 in a loop preceding strand 3, and Gly167-Thr169 ($r_{4–6}$) in helix 4 (Figs 1c and 6a). Most of these residues are conserved or replaced by similar amino acid residues in PfGrx or other enzymatically active glutaredoxins. In contrast, iron–sulfur cluster-binding glutaredoxins such as ScGrx3-5 have a modified loop and helix 3 (Fig. 1c)[3]. Some of these modifications might stabilize the interaction between the glutathione moiety at the scaffold site and the iron–sulfur cluster. In the absence of the cluster, such monothiol glutaredoxins can still be glutathionylated[26,27,29] and might (slowly) transfer the glutathione moiety to reduced cysteine residues as suggested for the AtGrxS15-dependent oxidation of reduced roGFP2 by GSSG. Comparing the glutathione-scaffold site among a variety of other members of the thioredoxin superfamily also provides an explanation for their disulfide substrate preferences, for example, why glutaredoxins prefer GSSR in contrast to thioredoxins lacking most of the relevant residues (Fig. 1c). Hence, the assignment of the glutathione-scaffold site could facilitate the prediction of disulfide substrate preferences of uncharacterized members of the thioredoxin superfamily.

A structure-based assignment of the glutathione activator site without transition-state analogues is much more difficult because of labile GSH interactions[2] that likely depend not only on the presence of the first glutathione moiety but also on conformational changes of the glutathionylated enzyme[3,12,37,44]. For example, NMR-titration experiments of non-glutathionylated poplar GrxC4 or ScGrx8 with GSH required high glutathione concentrations to detect chemical shifts[47,48] (and titrations of active site serine mutants with GSSG cannot discriminate between both glutathionyl moieties). Here we identified Lys105 of ScGrx7 and Lys26 of PfGrx as a dual activator for the catalytic cysteine residue and GSH. A conserved dual function of the conserved lysine residue for redox catalysis is also supported by previous studies: Lys19 of human Grx1 and Lys8 of NrdH-redoxin from *C. glutamicum* were shown to contribute to the low $pK_a$ value of the active site cysteine[37,38]. Furthermore, in agreement with our data on ScGrx7 and PfGrx, Jao *et al.*[37] reported decreased $k_{cat}^{app}$ and increased $K_m^{app}$(GSSCys) values upon replacement of Lys19 in a single cysteine mutant of human Grx1. Even though variable GSH concentrations were not tested, the authors suggested that Lys19 replacements have a negative effect on the reactivity of the glutathionylated enzyme with the reducing agent. Replacement of Lys8 in oxidized *E. coli* Grx3 altered the equilibration kinetics with reduced thioredoxin 1.

Shekther *et al.*[49] therefore suggested an important role of the residue as a gatekeeper to modify the reactivity of reducing or alkylating agents. Furthermore, ScGrx8, which has an alanine residue at the same position (Fig. 1c), was shown to have a very low activity in standard assays[12,48]. Please note that Lys105 and homologous residues adopt quite similar positions in the structures of active and inactive glutaredoxins[3]. Thus, the presence of the basic residue seems to be a necessary but not a sufficient condition for GSH activation. Regarding additional residues of ScGrx7 that could form an interaction site for GSH, we suggest Asp144 and Glu147 in helix 3 as well as the hydroxyl group of conserved Tyr110: First, helix 3 and Tyr110 are both replaced in enzymatically inactive glutaredoxins[11,12] (Fig. 1c) and preliminary mutational studies on Tyr110 show indeed similar effects to Lys105 (Supplementary Figs 18 and 19). Second, Lys105, Tyr110, Asp144 and Glu147 all stick out on the protein surface, are on top of the scaffold site and form a protein area in a 120° angle to the glutathione-scaffold site (Fig. 6a). This arrangement presumably prevents the electrostatic repulsion of the two glutathione moieties on the surface of active glutaredoxins. In contrast, helix 3 in iron–sulfur cluster-binding monothiol glutaredoxins contains a bulky tryptophan residue in a WP-motif that sticks out on the protein surface[3]. These and other modifications presumably prevent an efficient interaction with GSH as reflected by the absent GSH-dependent reduction of oxidized roGFP2 in the presence of AtGrxS15. In summary, alterations at the glutathione activator site result in an absent geometric and electrostatic complementary[3,50] for the reducing agent, explaining the inefficient reduction of numerous glutaredoxins[12,25–29,48] and thioredoxins by GSH. As a consequence, such inactive redox proteins are kinetically uncoupled from the GSH pool. The physiological relevance of the kinetic uncoupling is that it allows the formation of stable complexes, for example, with iron–sulfur clusters or disulfide-bonded interaction partners. Future studies will reveal the applicability of our model, for example, to activate enzymatically inactive glutaredoxins, to alter the sensitivity and responsiveness of glutaredoxin-coupled fluorescent redox sensors, to further address the catalytic mechanism with GSSCys, HEDS and physiological substrates, or to design specific transition-state inhibitors.

Which part of the catalytic cycle is rate-limiting for glutaredoxin catalysis? ScGrx7 and PfGrx both have typical ping-pong kinetics with GSSCys and GSH and cannot be saturated at infinite substrate concentrations in accordance with previous results from GSSEtOH assays on ScGrx7 (ref. 18). Infinite $k_{cat}$ values were also reported for human dithiol glutaredoxins as well as poplar GrxS12 and are presumably a common feature of enzymatically active glutaredoxins[2,10,20,21] and glutathione-dependent hydroperoxidases[33,51]. Thus, neither $k_2$ nor $k_5$ in Fig. 2f is rate-limiting and no enzyme–substrate complexes accumulate in contrast to enzymes with typical Michaelis–Menten kinetics[2,39,52]. An absent inhibition of human glutaredoxins by *S*-methylglutathione furthermore suggests that the enzymes react in an encounter reaction[2,19,20]. Short-lived enzyme–substrate interactions without stable complex formation are also in accordance with infinite $k_{cat}$ values and an inefficient competitive inhibition of ScGrx7. Nevertheless, the kinetic data for our mutants and rate constants for ScGrx7 and PfGrx between $10^5$–$10^6$ M$^{-1}$ s$^{-1}$ are too slow for a diffusion-controlled reaction and point towards important short-lived enzyme–substrate interactions and rate-limiting re-orientations of the substrate at the enzyme surface. Reduction of ScGrx7 by GSH is slower than the oxidation by GSSCys (unless Lys105 is replaced, which decelerates the oxidative half-reaction even more than the reductive half-reaction owing to the dual activator

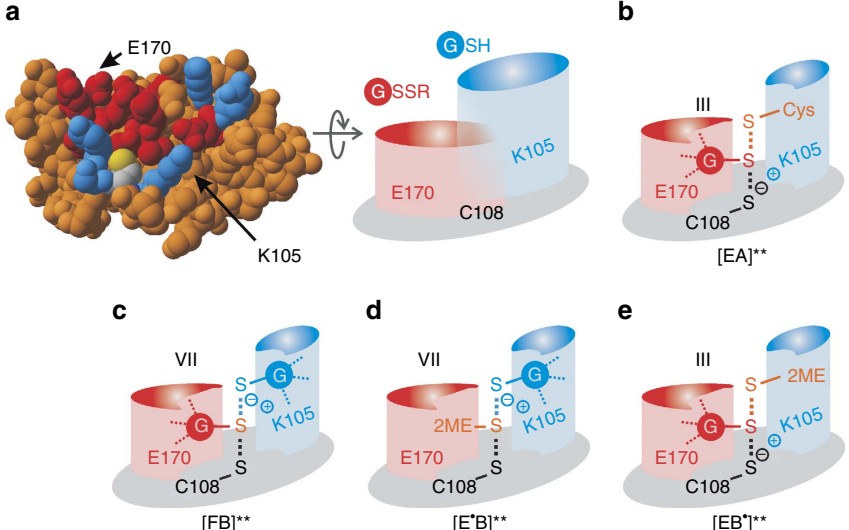

**Figure 6 | Model of the glutathione scaffold and glutathione activator site with implications for relevant transition states.** (**a**) Residues of the glutathione-scaffold site (including confirmed Glu170) and the potential glutathione activator site (including confirmed Lys105) both face the active site cysteine residue and are highlighted in red and blue, respectively. A schematic representation of both sites at the enzyme surface is shown on the right side of the panel. (**b–e**) Proposed transition states for the GSH-dependent turnover of the model substrates GSSCys and HEDS. Transition states [EA]** with GSSCys in **b** and [EB•]** with GSSEtOH in **e** both correspond to species III of the original glutathione-scaffold model[12] and yield the glutathionylated enzyme 'F'. Transition state [FB]** with GSSG in **c** and the HEDS-derived transition state [E•B]** with GSSEtOH in **d** correspond to species VII of the original activator model[12] with a glutathione and a non-glutathione disulfide substrate, respectively. Please note that the glutathionylated enzyme 'F' cannot be directly formed from [E•B]** in **d** because of geometric constraints that prevent a direct attack of GS⁻ at the active site cysteine. The product GSSEtOH first has to change its orientation, as shown in **e**, before it serves as a substrate yielding glutathionylated enzyme 'F'.

function). The difference is even more pronounced for PfGrx$^{C32S/C88S}$. Its reductive half-reaction with GSH has a rate constant of $1.2 \times 10^5$ $M^{-1}s^{-1}$, which is ten times slower than the oxidative half-reaction with GSSCys. Rate constants for mammalian glutaredoxins and poplar GrxS12 range from $2.5 \times 10^4$ to $2 \times 10^6$ $M^{-1}s^{-1}$ and also support a rate-limiting reductive half-reaction[19,20,21,37]. Thus, for wild-type glutaredoxins, formation of the transition state between the reduced enzyme and GSSCys (Fig. 6b) is more efficient than the formation of the transition state between the glutathionylated enzyme and GSH (Fig. 6c). A plausible explanation could be a reorientation of Lys105 towards GS⁻ resulting in a decreased stabilization of the active site thiolate as a leaving group. The predicted transition states for HEDS catalysis suggest that GSSEtOH binds in two different orientations because it is a product and substrate (Fig. 6d,e). Only the substrate orientation (Fig. 6e) is productive regarding the formation of glutathionylated enzyme. This might explain the common x axis intercept in Lineweaver–Burk plots[11,17,18], which resemble a non-competitive inhibition pattern with identical dissociation constants for the inhibitor and substrate[52].

In conclusion, we revealed novel structure–function relationships of glutaredoxins, gained insights regarding the enzymatic conversion of glutathione- and non-glutathione disulfide substrates, and identified two distinct substrate interaction sites that include a scaffold residue and the conserved dual activator Lys105 in ScGrx7 and Lys26 in PfGrx. Our study has important implications for our understanding of enzymatically active and inactive glutaredoxins and could be useful for the design and optimization of artificial glutaredoxins, glutaredoxin-coupled fluorescent redox sensors and transition-state inhibitors.

## Methods

**Materials.** GSH, GSSG, 2-mercaptoethanol, ophthalmic acid, S-methylglutathione and yeast glutathione reductase (GR) were obtained from Sigma-Aldrich, HEDS was from Alfa Aesar, GSSCys from Toronto Research Chemicals and NADPH was from Gerbu. PCR primers were purchased from Metabion. N terminally MRGS(H)$_6$-tagged wild type and mutant ScGrx7, PfGrx and PfGR were expressed in *E. coli* strain XL1-Blue and purified by affinity chromatography using an elution buffer containing 200 mM imidazole, 300 mM NaCl and 50 mM sodium phosphate, pH 8.0 (refs 11,12,18,33,53). Recombinant His-tagged AtGrxS15, AtGrxC1 and roGFP2 were also expressed and purified according to established protocols[34,36].

**Site-directed mutagenesis.** Point mutations were introduced by PCR with *Pfu* polymerase (Promega) using the double stop-codon construct of pQE30/SCGRX7 (ref. 12) as a template and the mutagenesis primers listed in Supplementary Table 9. Likewise, mutants of pQE30/PFGRX$^{C32S/C88S}$ were generated with the primers listed in Supplementary Table 10. Following the digestion of the methylated template DNA by *Dpn*I (NEB), plasmids were transformed into competent *E. coli* XL1-Blue cells. Correct mutations and sequences were confirmed by sequencing both strands.

**Structure visualization and residue selection.** Protein structures of glutaredoxins were inspected using Swiss-PDB Viewer[54]. Electrostatic potentials for ScGrx6 and ScGrx7 were computed and mapped to the protein surface using the Poisson–Boltzmann method of Swiss-PDB Viewer (dielectric constant solvent: 80, atomic partial charges, dielectric constant protein: 4, solvent ionic strength: 0.1 M).

**GSSCys and HEDS oxidoreductase assays.** Steady-state kinetics of wild type and mutant ScGrx7 and PfGrx in the GSSCys and HEDS assays were determined spectrophotometrically by monitoring the consumption of NADPH at 340 nm and 25 °C using a thermostated Jasco V-650 UV/vis spectrophotometer[11,12,18]. Fresh stock solutions of NADPH, GSH, GR and GSSCys or HEDS were prepared in assay buffer containing 0.1 M Tris/HCl, 1 mM EDTA, pH 8.0 before each experiment. Both assays were performed with 0.1 mM NADPH and 1 U ml$^{-1}$ GR. Final concentrations of K105X and E170X mutants were 10 nM to 1.5 µM and 5–20 nM, respectively. Final concentrations of DM, D90A and K26A were 5–10 nM, 5–12.5 nM and 50–100 nM, respectively. For the GSSCys assays, either GSH was varied between 50 µM and 2.0 mM at fixed GSSCys concentrations (25, 50, 100 and 150 µM) or GSSCys was varied between 25 and 200 µM at fixed concentrations of GSH (0.05, 0.1, 0.2, 0.3 and 1.0 mM for ScGrx7 or 0.3, 0.5, 1.0 and 2.0 mM for PfGrx). NADPH, GSH and GR were mixed in assay buffer before Grx was added and a baseline was recorded for 30 s. All GSSCys assays were started by the addition of GSSCys. For the HEDS assays, either GSH was varied between 100 µM and 1.5 mM for ScGrx (or 250 µM and 3.0 mM for PfGrx) at fixed concentrations of

HEDS (0.18, 0.37, 0.55 and 0.74 mM) or HEDS was varied between 90 and 920 µM at fixed concentrations of GSH (0.3, 0.5, 1 and 1.5 mM for ScGrx7 or 0.25, 0.5, 1 and 2 mM for PfGrx). NADPH, GSH and HEDS were preincubated in assay buffer for 2 min before GR was added and a baseline was recorded for 30 s. All HEDS assays were started by the addition of enzyme. Kinetic data were analysed in Excel and SigmaPlot 12 by non-linear and linear regression according to Michaelis–Menten, Lineweaver–Burk, Eadie–Hofstee and Hanes theory[11,12,18].

**Grx-catalysed redox reactions of roGFP2.** Redox reactions of roGFP2 in the presence of AtGrxS15 were analysed *in vitro*[34–36]. Ratiometric time-course measurements were carried out with initially oxidized or reduced roGFP2 on a BMG POLARstar Omega fluorescence plate reader with a filter-based excitation at 390 ± 10 and 480 ± 10 nm. Emitted light was detected at 520 nm with a bandwidth of 10 nm. Samples in 96-well plates contained 0.1 M potassium phosphate buffer, pH 7.8, 1 µM roGFP2 and 1, 3 or 10 µM AtGrxS15. Samples for measurements with GSH were also supplemented with 1 U GR and 100 µM NADPH to remove traces of GSSG and to maintain a highly reducing redox state. Reduced roGFP2 was obtained after a 20 min treatment with 10 mM DTT, which was subsequently removed on a ZebaTM Spin Desalting Column (Thermo Scientific). For interaction analysis with oxidized roGFP2, GSH (in 0.1 M potassium phosphate buffer, pH 7.0) was injected into the wells to a final concentration of 2 mM. For interaction analysis with reduced roGFP2, GSSG was injected into the wells to a final concentration of 40 µM. Fully oxidized and reduced states of roGFP2 with maximum and minimum fluorescence ratios at 390/480 nm were determined with 10 mM $H_2O_2$ and 10 mM DTT, respectively. A basal background fluorescence of buffer or buffer containing 100 µM NADPH was subtracted from the samples. The redox kinetics of roGFP2 in the presence of AtGrxC1 served as a positive control.

**Data availability.** The authors declare that data supporting the findings of this study are available within the article and its Supplementary Information files and from the corresponding author upon reasonable request. The PDB structures with the following accession codes 2WCI, 3L4N, 3D4M, 3D5J, 2M80, 2WUL, 2WOU, 1MEK, 1B4Q and 4FIW were used in this work.

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

## Acknowledgements

P.B. was supported by the HBIGS MD fellowship programme. M.D. acknowledges the Deutsche Forschungsgemeinschaft (DFG) for funding of his own position in the frame of the Heisenberg programme (Grant DE 1431/9-1) and for funding L.L. (Grant DE 1431/10-1). This work was partially funded by the DFG priority programme SPP 1710 (Grant DE 1431/8-1 to M.D. and Grant ME 1567/9-1 to A.J.M.). We furthermore thank Luise Krauth-Siegel for helpful discussions and Tobias Dick and Bruce Morgan for reading an initial version of the manuscript and suggestions.

## Author contributions

P.B. performed all kinetic measurements with ScGrx7. L.L. generated the point mutants of ScGrx7 and PfGrx, and performed all kinetic measurements with PfGrx. P.B., L.L. and M.D. analysed the kinetic data and prepared the figures. M.D. conceived the experiments, supervised the study and wrote the manuscript. A.M. and A.J.M. conceived, performed and analysed the roGFP2 experiments. All authors discussed the results and gave approval to the final version of the manuscript.

## Additional information

**Competing interests:** The authors declare no competing financial interests.

