## [Peer Review File · Nature Communications]

Reviewer #1 (Remarks to the Author)

This is a carefully designed and executed study. The manuscript is well written and logically presented in a systematic fashion. The study sets out to document a conceptual framework for understanding the mechanism(s) of catalysis by glutaredoxin enzymes that was originally described and illustrated in an excellent review (ref. #3) by the communicating author of this manuscript who displays a comprehensive command of the glutaredoxin literature. Kinetic studies are focused on WT and site-specific mutant (K105X and E170X) forms of the yeast enzyme ScGrx7 which was chosen as a prototype because it has single cysteine residue, simplifying the kinetics. Although the conclusions are logical and consistent with the kinetic data, there are experiments that would provide more direct evidence for the proposed mechanisms. It seems problematic to define separate interaction sites for the glutathionyl tripeptide moiety by mutating only one residue at each putative site. Some of the differences reported are small in magnitude and the lack of statistical analyses makes decisions about the significance of these changes challenging. Also, by using a single form of the glutaredoxin family of enzymes it is difficult to make a convincing argument about the generality of the conclusions. Overall the premise for the study is very well set up, but the scope of the data seems limited. Specific comments/questions are delineated below.

1. Intro, p. 4, ls.88-90, "ScGrx7 is particularly suited for unbiased kinetic and mechanistic studies because of its high activity, the absence of iron-sulfur clusters in enzyme preparations, and the lack of additional cysteine residues" / Comment: There are other active monothiol forms of Grx that could be tested also; and dithiol forms Grx in which the second cysteine is mutated to serine (or alanine) would serve. By including other Grx forms the generality of the interpretations could be reinforced.

2. Re: Figure 1c: Why are no mammalian forms of glutaredoxin included in this comparison? For example, NMR structures of the Grx-SSG derivatives of EcGrx1 (ref. #14) (included in Fig. 1c) and hGrx1 (ref. # 31) (not included) identify analogous residues in each that stabilize the bound glutathionyl moiety (i.e., the "GSH scaffold site," according to the nomenclature of this manuscript).

3. Intro, p. 4, ls.96-100, "The requirement of both sites for the efficient reduction of glutathionylated disulfide substrates by GSH explains the deviating properties and substrate preferences of enzymatically active and inactive glutaredoxins as well as thioredoxins with implications for the design and optimization of artificial enzymes and inhibitors." / This seems to be a quite expansive statement based only on studies of mutations of 2 residues in a single enzyme form.

4. p. 7, l.170-174: "For example, the tenfold lower K_m app (GSH) of K105E did not result from a higher affinity for GS- (this mutation should have actually decreased the affinity for the negatively charged substrate) but from lower steady-state concentrations of the glutathionylated enzyme 'F', yielding apparent substrate saturations at lower GSH concentrations (Fig. 2b)." /Comment: This is a very good insight and an important distinction. However, readers who are not conversant with enzyme mechanisms beyond the basic fundamentals may not appreciate this important distinction.

5. p. 7, ls. 179-181, "The drastic deceleration of the oxidative half-reaction of K105X mutants furthermore suggests that the residue also stabilizes the enzyme thiolate form 'E'." / Comment: This conclusion could be substantiated by determining the pKa value for the active site cysteine residue in the WT and K105X mutant forms.

6. p. 9, ls. 233-234, "In summary, we propose a catalytic model for the HEDS assay with two different covalent enzyme modifications, GSSEtOH as a product and substrate, and Lys105 as a glutaredoxin and glutathione activator." Comment: This description seems to suggest a multistep on-enzyme processive reaction by which HEDS reacts with Grx to generate Grx-SSEtOH with

release of only 2ME; then GSH reacts with the first intermediate to generate GSSEtOH on the enzyme; then GSSEtOH reacts with the active cysteine to form of Grx-SSG, which reacts with GSH to form GSSG and regenerate the free enzyme. This is a complex reaction scheme which would seem to be inefficient if the GSSEtOH product was released and had to re-associate as a substrate from solution.

Are there other data to support this proposed mechanism? For example, if the first reaction of the Grx enzyme with HEDS is formation of the Grx-SSEtOH intermediate as suggested, then a stoichiometric burst of 2ME should occur in the absence of GSH, and the Grx-SSEtOH intermediate could be isolated and documented by mass spectrometry. For dithiol Grx enzymes the burst of 2ME should equal two times the Grx concentration, and the intramolecular disulfide form Grx(S₂) could be isolated and documented.

7. p. 9, ls. 242-3/ re E170X mutants, "Effects for the E170X mutants on the determined kinetic parameters were in general less pronounced than for the K105X mutants." Comment: This appears to be a serious understatement. Scrutiny of Supplementary Table 1 indicates > order of magnitude differences for the K105 mutants; but the E170 differences show at most a factor of two difference. Furthermore there are no errors indicated for these estimates, and therefore no tests for significant differences.

Corroborating data for the role of E170 in the "GSH scaffold site" could be obtained by testing the specificity of the enzyme for CysSSR substrates where R is other than glutathione. If decreased glutathionyl specificity were observed for the E170X mutants, this result would contribute more directly to the interpretation that E170 is an important component of the GSH scaffold site.

8. p. 10, ls. 275-6, "Regression and pattern analyses revealed sequential kinetics for all mutants with an apparent common intersection point at the x-axis (Supplementary Fig. 8) / Comment: Is the sequence ordered? GSH or HEDS first? or is it random?

9. p. 11, ls. 283-4, "Mutation of Glu170 decreased the reciprocal Dalziel coefficient θ_1 by up to 56% whereas θ_2 remained constant (Fig. 5d). Comment: In the case of the HEDS reaction, what is the step referred to by θ_1 , Grx-SSEtOH formation or Grx-SSG formation? How would this be distinguished? (See next question)

10. p.11, 289-91, "The decrease of the $1/\Phi_1$ values (Fig. 5d) therefore suggests a less efficient interaction between E170X mutants and HEDS-derived GSSEtOH in accordance with the scaffold model and the hypothesis that GSSEtOH formation is catalyzed by ScGrx7." /Comment: There is concern that these kinetics (especially for HEDS) may be peculiar to this form of GRx, precluding broad application of the interpretations. At least one more active form of GRx should be tested in order to validate the proposed catalytic mechanisms.

11. p. 12, ls. 331-332 "Furthermore, the assignment of the glutathione scaffold site could facilitate the pre-diction of disulfide substrate preferences of uncharacterized members of the thioredoxin family." / Comment: How can a scaffold site be identified by a single residue? What other residues are important to this interaction? (see next comment where multiple residues are suggested for the GSH activator site besides K105).

12. p. 13, ls. 350-352, "Regarding additional residues of ScGrx7 that could form an interaction site for GSH, we suggest Asp144 and Glu147 in helix 3 as well as the hydroxyl group of conserved Tyr110/ Comment: Although no mutagenesis studies are provided to support these interpretations, still this discussion is a better attempt to describe an activation motif than the assignment of a single residue as the scaffold site.

13. Methods, p. 20, l. 544, "All assays were started by addition of enzyme." Comment: Does this mean that substrates were pre-incubated? If so, for how long? It is important to describe the time

allotted for temperature equilibration, especially for the HEDS reaction where non-enzymatic formation of GSSEtOH would occur and affect the interpretation of the reaction mechanism.

14. Re: supplementary figures/ What were the criteria for omitting the values shown in parentheses from the regression analyses? How were the lines fit - linear regression of the double reciprocal values, or non-linear fit to rectangular hyperbolic relationships? These details should be included.

15. Re: Supplementary Tables/ There should be more complete legends for the supplementary tables. In Supplementary Tables 2-6, how are the errors determined? These errors should be included in the values of "percent relative to wild type," which should be defined in the legend. Importantly, significant differences should be indicated.

Reviewer #2 (Remarks to the Author)

The paper entitled "Glutaredoxin catalysis requires two distinct glutathione interaction sites" by Patricia Begas and collaborator describes a thorough kinetic study of a yeast glutaredoxin, namely GRX7. They have analyzed under steady-state conditions the kinetic parameters of an intact recombinant protein and of a certain number of variants mutated for two residues (Lys105 and Glu170) with two different substrates thought to be representative of glutathionylated substrates (L-cysteine-glutathione disulfide, GSSEtOH) or of non-glutathione substrates (bis(2-hydroxyethyl)disulfide, HED). The aim was to experimentally validate or invalidate two previously proposed hypothetical models for glutaredoxin catalysis, which basically assumed the existence of two distinct protein areas, one for the recognition of the disulfide substrate (scaffold site) and one for the recognition of the reducing glutathione (activator site). These two residues have been selected based on a structural amino acid alignment and on their conservation in glutaredoxins which exhibit thiol disulfide exchange reactions compared to those which do not generally exhibit GSH-dependent activity at least with HED.

The enzymatic experiments seem to have been very carefully conducted. Considering the very specialized expertise needed to understand the data, they are quite clearly presented with appropriate statistics.

The interpretation of the kinetic data made by the authors suggests that there are indeed two successive steps for recognizing glutathione moieties involving at least two different residues, Glu107 being part of the scaffold site whereas Lys105 is rather part of the activator site. Other residues have been proposed to participate to these respective sites, but there is no associated experimental demonstration.

If these observations turn to be true for other residues and other enzymes, this would constitute a major advance in our understanding of the biochemical properties of the different Grx subclasses. In the absence of generalization, this might be my major criticism/concern. Does this model really explain differences between what has been defined as "active and inactive" glutaredoxins.

I have a relatively few minor other concerns detailed below:

- In the abstract (two times) and at many places, the authors use the definition "enzymatically active and inactive glutaredoxins". This is an incorrect shortcut because as rightly defined once in the introduction, this definition essentially applies to the glutathione-dependent reactions and even more specifically with HED as a substrate. These presumed inactive glutaredoxins may in fact have specific substrates and reducing pathways as shown or suggested by a few studies.

- Lines 114-118: "In accordance with the glutathione activator hypothesis, only positively charged amino acids replace Lys105 in enzymatically active glutaredoxins, whereas in inactive isoforms and other members of the thioredoxin superfamily the activator candidate is often separated from the catalytic cysteine residue or is replaced by uncharged residues 3, 11, 12 (Fig. 1c). »

It is indicated that the amino acid alignment has been manually adjusted based on structural overlays and indeed this lysine is very conserved in Grxs and perfectly aligns in both "active and inactive" isoforms. So, does it mean that the Lys is at the right or same position in inactive GRXs although separated from the catalytic Cys by a few residues? This has to be carefully verified from available structures.

- Line 126-127 "...might be sandwiched between the protein and the glutathione moiety of GSH". There should be something wrong here.

- While the Lys105 is extremely conserved, Glu170 is less conserved although the amino acid sequences used for the alignment have been judiciously chosen. Moreover the proposal that Asp144, Glu147 and Tyr110 could be part of the "scaffold" interaction site with GSH is more speculative in the absence of experimental support and of 3D structure for ScGrx7 and considering that Asp144 and Glu147 are relatively weakly conserved and that Tyr110 is replaced by a Phe in most if not all "inactive" Grxs but this Phe is found also in an important set of "active" Grxs. Considering these observations and in the absence of further kinetic and structural data and since the authors validated their model of "scaffold and activator sites" only for one residue, I would be more cautious with the description of the glutaredoxin active site as two distinct glutathione interaction sites. In fact, this can also be presented as the successive involvement of specific residues during the catalytic act.

Other comments:

- Line 353: "Regarding additional residues of ScGrx7 that could form an interaction site for GSH, we suggest Asp144 and Glu147 in helix 3 as well as the hydroxyl group of conserved Tyr110: First, helix 3 and Tyr110 are both replaced in enzymatically inactive glutaredoxins^{11,12} (Fig. 1c). Second, the four residues stick out on the protein surface, are on top of the scaffold site".

I guess the number four refers to these three residues and to Glu170 but this is not clear as it presently states.

-Fig 1b: According to the legend and unless I am missing something, I would have expected to see a GSSR and not a GSSG in the "glutathione scaffold model" part.

Reviewer #3 (Remarks to the Author)

The manuscript 'Glutaredoxin catalysis requires two distinct glutathione interaction sites' by M. Deponte and co-workers addresses the basis for the reduction of the model substrates HEDS and Cys-S-SG by glutaredoxins with one molecule of GSH. The authors constructed two mutants and analysed the steady state kinetics of these proteins to an unparalleled extend. Based on these analysis, the authors concluded two distinct glutathione interactions sites. First, the site that interacts with the glutathione moiety of glutathionylated disulfide substrates, and secondly, the site that interacts with glutathione as the reducing agent of the intermediate mixed disulfide between the glutaredoxin and the first glutathione.

Glutaredoxins, the structurally and functionally most diverse group of the thioredoxin fold family, possess fundamental roles in metabolic electron supply, cell signalling, and iron metabolism. Many of the proteins are essential for the development or viability of the respective organisms. Hence, the topic of the manuscript may be of interest to a general audience. In fact, I am convinced that this study will also be appreciated by a wide audience and may thus be suitable for publication in Nature Communications.

Based on their in-depth and in many aspects innovative kinetic analysis, the authors in deed provide the first experimental evidence for two distinct glutathione (moiety) interaction sites for

both the ping and the pong reactions of the glutaredoxin reactions cycle.

While the first binding site was well established before, primarily based on solved structures of GSH-Grx complexes, the existence of the second interaction site is the primary novelty of this study.

These findings solve the paradox of glutathione binding despite the lack of any reactivity, as in CGFS-type glutaredoxins. The conclusions drawn are comprehensible, the discussion is balanced and appropriately critical. In fact, a brief structural comparison of different structures of GSH-glutaredoxin complexes (E. c. Grx3 and H.s. Grx2) confirmed that GSH in FeS-Grx/GSH complexes and in mixed disulfides (subsequent to the first half-reaction) occupy the same binding site.

Undoubtedly, these results represent a major step in our understanding of the diverse functions of glutaredoxins in redox biochemistry and metal homeostasis.

Suggested improvements:

In their study, the authors demonstrated the importance of E170 (S.c. Grx7 numbering) for the first interaction site. The authors also discuss the potential involvement of other residues and side chains for this interaction. It would be extremely valuable if the authors would put some more of these residues to the test, especially the hydroxyl group of the active site Y110.

It would be helpful if the authors also include a brief discussion on the impact of their findings for our understanding of iron-sulfur ligation by glutaredoxins in concert with glutathione

Some details on the statistical analysis of the results from the kinetic measurements are requested to be able to better judge the robustness of the conclusions drawn.

Comments to Referees

Reviewer #1 (Remarks to the Author):

This is a carefully designed and executed study. The manuscript is well written and logically presented in a systematic fashion. The study sets out to document a conceptual framework for understanding the mechanism(s) of catalysis by glutaredoxin enzymes that was originally described and illustrated in an excellent review (ref. #3) by the communicating author of this manuscript who displays a comprehensive command of the glutaredoxin literature. Kinetic studies are focused on WT and site-specific mutant (K105X and E170X) forms of the yeast enzyme ScGrx7 which was chosen as a prototype because it has single cysteine residue, simplifying the kinetics. Although the conclusions are logical and consistent with the kinetic data, there are experiments that would provide more direct evidence for the proposed mechanisms. It seems problematic to define separate interaction sites for the glutathionyl tripeptide moiety by mutating only one residue at each putative site. Some of the differences reported are small in magnitude and the lack of statistical analyses makes decisions about the significance of these changes challenging. Also, by using a single form of the glutaredoxin family of enzymes it is difficult to make a convincing argument about the generality of the conclusions. Overall the premise for the study is very well set up, but the scope of the data seems limited. Specific comments/questions are delineated below.

Reply: We thank the referee for his careful review and constructive criticism.

1) Additional kinetic data on the homologous residues in the non-related enzyme PfGrx from malaria parasites are now included in the updated manuscript. 2) Furthermore, we also analyzed an inactive monothiol glutaredoxin from Arabidopsis in collaboration with Andreas Meyer revealing that the oxidative half-reaction with GSSG is functional while the reductive half-reaction with GSH is inactivated. Both data sets are included as Supplementary Fig. 11-17 and Supplementary Tables 6-8 and are presented and discussed in the corresponding paragraphs of the revised manuscript. 3) We also performed a preliminary mutational study on Tyr110 as a third residue of ScGrx7 revealing similar patterns as for Lys105 as predicted. We did not include these confidential data on Tyr110 because we need more replicates and mutants to confirm the findings. Furthermore, we think that an exact mapping of the active site goes beyond the scope of this manuscript (which already includes an extreme amount of kinetic data). All new data sets are in agreement with our previous measurements and the proposed catalytic model. 4) We also added statistical analyses on all apparent k_{cat} and K_m values as requested. The obtained P-values support our statements even for smaller differences among mutants. The P-values are summarized in the separate supplementary Table 11 (10 pages) to avoid a crowding of symbols that would render the bar charts unreadable.

1. Intro, p. 4, ls.88-90, "ScGrx7 is particularly suited for unbiased kinetic and mechanistic studies because of its high activity, the absence of iron-sulfur clusters in enzyme preparations, and the lack of additional cysteine residues" / Comment: There are other active monothiol forms of Grx that could be tested also; and dithiol forms Grx in which the second cysteine is mutated to serine (or alanine) would serve. By including other Grx forms the generality of the interpretations could be reinforced.

Reply 1.1: In order to support our general statements and to address the valid concerns, we analyzed the mutant PfGrx^{C32S/C88S} of the dithiol Grx from Plasmodium falciparum, which we characterized previously (Djuika et al. 2013 BBA & Begas et al. 2015 Chem Science). First, we generated Ala-mutants of the corresponding residues Lys26 and Asp90 in PfGrx and subsequently determined the steady-state kinetics with HEDS and GSSG. The obtained effects are summarized in Fig. 11-16 and Supplementary Tables 6-8 and are similar to ScGrx7. We also analyzed GRXS15 from Arabidopsis using reduced and oxidized roGFP2 as a substrate. These studies support our hypothesis that the

oxidative half-reaction of such 'inactive' monothiol glutaredoxins is functional in contrast to the reductive half-reaction with GSH. In summary, our two-site model is now supported by studies on three non-related glutaredoxins.

2. Re: Figure 1c: Why are no mammalian forms of glutaredoxin included in this comparison? For example, NMR structures of the Grx-SSG derivatives of EcGrx1 (ref. #14) (included in Fig. 1c) and hGrx1 (ref. # 31) (not included) identify analogous residues in each that stabilize the bound glutathionyl moiety (i.e., the "GSH scaffold site," according to the nomenclature of this manuscript).

Reply 1.2: We added the sequence of HsGrx1 below HsGrx2 to the alignment of Fig. 1c as requested and modified the figure legend for clarity (PDB entries 1B4Q and 1GRX were added and abbreviations for *Homo sapiens*, yeast etc. were introduced). Please note that the original alignment already contained the sequence of mammalian (*Homo sapiens*) HsGrx5 and HsGrx2 with the relevant residues highlighted for comparison.

3. Intro, p. 4, ls.96-100, "The requirement of both sites for the efficient reduction of glutathionylated disulfide substrates by GSH explains the deviating properties and substrate preferences of enzymatically active and inactive glutaredoxins as well as thioredoxins with implications for the design and optimization of artificial enzymes and inhibitors." / This seems to be a quite expansive statement based only on studies of mutations of 2 residues in a single enzyme form.

Reply 1.3: Our two-site model is now supported by studies on three non-related glutaredoxins as outlined above. We hope that these experiments justify the general statements.

4. p. 7, l.170-174: "For example, the tenfold lower K_m app (GSH) of K105E did not result from a higher affinity for GS- (this mutation should have actually decreased the affinity for the negatively charged substrate) but from lower steady-state concentrations of the glutathionylated enzyme 'F', yielding apparent substrate saturations at lower GSH concentrations (Fig. 2b)." /Comment: This is a very good insight and an important distinction. However, readers who are not conversant with enzyme mechanisms beyond the basic fundamentals may not appreciate this important distinction.

Reply 1.4: We added the following sentence and made the following modification to clarify and emphasize the statement. "In other words, $K_{mapp}(GSSCys)$ and $K_{mapp}(GSH)$ are not solely defined by the ratios k_{-1}/k_1 and k_{-4}/k_4 but are also affected by other rate constants in Fig. 2f. ...(this mutation should have actually decreased the affinity and k_4/k_{-4} ratio for the negatively charged substrate)..."

5. p. 7, ls. 179-181, "The drastic deceleration of the oxidative half-reaction of K105X mutants furthermore suggests that the residue also stabilizes the enzyme thiolate form 'E'." / Comment: This conclusion could be substantiated by determining the pKa value for the active site cysteine residue in the WT and K105X mutant forms.

Reply 1.5: We agree that pKa values would support our interpretation but decided to address this aspect in a follow-up study because it does not directly affect the major question whether there are two distinct glutathione interaction sites or not. Please take into account that the manuscript already contains numerous data sets and that a thorough determination and interpretation of cysteine pKa values would require several additional time-consuming experiments.

6. p. 9, ls. 233-234, "In summary, we propose a catalytic model for the HEDS assay with two different covalent enzyme modifications, GSSEtOH as a product and substrate, and Lys105 as a glutaredoxin and glutathione activator." Comment: This description seems to suggest a multistep on-enzyme processive reaction by which HEDS reacts with Grx to generate Grx-SSEtOH with release of only 2ME;

then GSH reacts with the first intermediate to generate GSSEtOH on the enzyme; then GSSEtOH reacts with the active cysteine to form of Grx-SSEtOH, which reacts with GSH to form GSSG and regenerate the free enzyme. This is a complex reaction scheme which would seem to be inefficient if the GSSEtOH product was released and had to re-associate as a substrate from solution.

Are there other data to support this proposed mechanism? For example, if the first reaction of the Grx enzyme with HEDS is formation of the Grx-SSEtOH intermediate as suggested, then a stoichiometric burst of 2ME should occur in the absence of GSH, and the Grx-SSEtOH intermediate could be isolated and documented by mass spectrometry. For dithiol Grx enzymes the burst of 2ME should equal two times the Grx concentration, and the intramolecular disulfide form Grx(S₂) could be isolated and documented.

Reply 1.6: We agree that the suggested scheme with a Grx-catalyzed generation and consumption of GSSEtOH is complex. The following key observations and experimental data support the mechanism: 1) We previously showed that the non-enzymatic reaction between GSH and HEDS is too slow to explain the high reaction rates in the HEDS assay for ScGrx7 and PfGrx^{C32S/C88S} (Begas et al. 2015 Chem Science). 2) The same study revealed that the relevance of the pre-incubation period between GSH and HEDS (yielding GSSEtOH) depends on the investigated enzyme system and that pre-incubation is far less relevant for the measured activity of ScGrx7 than expected (see also replies 13.1). 3) The kinetic patterns for ScGrx7 with HEDS that were determined in this as well as previous studies (Mesecke et al. 2008, Begas et al. 2015) revealed a constant apparent K_m value, which is typical for a non-competitive inhibition with identical dissociation constants for the inhibitor and substrate. As outlined in the discussion, this is expected in our mechanistic model because inhibitor and substrate (GSSEtOH) are identical.

The suggested experiment with HEDS and reduced enzyme in the absence of GSH will certainly result in the formation of Grx-SSEtOH and should be detectable by mass spectrometry. For example, Mieyal et al. 1991 showed that the human enzyme is protected from inactivation/alkylation by iodoacetamide when the enzyme is pretreated with HEDS or cystine. However, the suggested experiment will not tell us whether Grx-SSEtOH is just a thermodynamic endproduct or a kinetically relevant intermediate. This depends on the second order rate constant for the reaction between Grx and HEDS. To determine the rate constant and to discriminate between both scenarios one would need rapid kinetic experiments between Grx and HEDS in the presence and absence of GSH. Unfortunately, we do not have access to stopped-flow equipment in order to measure the initial formation/burst of 2-ME or to perform quenching experiments for the rapid detection of Grx-SSEtOH. We agree with the referee that future studies are necessary to fully decipher the reaction mechanism for the HEDS assay. However, such experiments go beyond the scope of this study, which was to show whether there are two different glutathione interaction sites or not. We rather consider the updated mechanistic model on the HEDS assay as a great bonus of the manuscript without putting too much emphasis on it.

7. p. 9, ls. 242-3/ re E170X mutants, "Effects for the E170X mutants on the determined kinetic parameters were in general less pronounced than for the K105X mutants." Comment: This appears to be a serious understatement. Scrutiny of Supplementary Table 1 indicates > order of magnitude differences for the K105 mutants; but the E170 differences show at most a factor of two difference. Furthermore there are no errors indicated for these estimates, and therefore no tests for significant differences.

Reply 1.7a: We agree that the effect of the E170 mutations is far less pronounced than for the K105 mutants and added 'far' to the sentence. The moderate effect makes sense considering that (i) several residues probably contribute to the GSSR interaction site, as exemplified by X-ray and NMR structures of glutathionylated glutaredoxins, and that (ii) the reductive half-reaction is overall rate-

limiting. Although the effect for the E170X mutants is less pronounced, the most important aspect is that a mutation of residue E170 alters the GSSR interaction but not the interaction with GSH. We now performed statistical analyses on all apparent k_{cat} and K_m values to further estimate the quality of our kinetic data. These analyses reveal, for example, that even smaller differences between WT enzyme & E170D on the one side and E170A & E170K on the other side have P-values < 0.01 or < 0.001 . The statistics have been added in a separate Supplementary Table 11 to avoid a crowding of symbols that would render the bar charts unreadable. Error bars are missing in the bar charts for the Dalziel coefficients because these are derived from the slopes of single secondary plots for each mutant. To convince the referee from the latter data quality, we added the r^2 values to the linear fits of the secondary plots as a statistical parameter to judge the differences between the Dalziel coefficients.

Corroborating data for the role of E170 in the "GSH scaffold site" could be obtained by testing the specificity of the enzyme for CysSSR substrates where R is other than glutathione. If decreased glutathionyl specificity were observed for the E170X mutants, this result would contribute more directly to the interpretation that E170 is an important component of the GSH scaffold site.

Reply 1.7b: A non-glutathione substrate is indeed a good approach to test the model, which is one reason why we conducted the assays with HEDS as a non-glutathione substrate. However, the problem with non-glutathione substrates such as CysSSR is that GSSR (or CysSSG) is formed in the assay and that the subsequent turnover of GSSR (or CysSSG) depends on E170 at the scaffold site. For example, we assume that similar kinetics would be observed for HEDS and cystine (Cys-SS-Cys) as non-glutathione substrates and that stopped-flow experiments will be necessary to decipher the complex mechanism for both non-glutathione substrates in future studies (see also replies 6.1 and 8.1). Another approach would be a more drastic alteration of the glutathione scaffold site by mapping the GSSR-interaction in further assays and combining mutations (resources and time provided). However, an exhaustive correlation between existing structural data on the scaffold site with kinetic data goes beyond the scope of this work. Our aim was to simply check whether there are two different glutathione interaction sites, which we think we convincingly demonstrated for ScGrx7 and now also for PfGrx.

8. p. 10, ls. 275-6, "Regression and pattern analyses revealed sequential kinetics for all mutants with an apparent common intersection point at the x-axis (Supplementary Fig. 8) / Comment: Is the sequence ordered? GSH or HEDS first? or is it random?"

Reply 1.8: As correctly summarized by referee 1 in his/her comment 6, we currently prefer a complex double ping-pong mechanism with GSSEtOH as a product and substrate that adopts two different orientations at the enzyme (Fig. 6d,e). Thus, HEDS is the first substrate, 2-ME#1 is the first product, GSH#1 is the second substrate, GSSEtOH is the second product, GSSEtOH with another orientation is the third substrate, 2-ME#2 is the third product, GSH#2 is the fourth substrate and GSSG is the fourth product. This could explain the non-competitive inhibition-like sequential kinetic patterns as mentioned in the discussion. Obviously, future studies are necessary to address the exact mechanism of non-glutathione disulfide substrates such as HEDS (see also replies 6.1 and 7.1).

9. p. 11, ls. 283-4, "Mutation of Glu170 decreased the reciprocal Dalziel coefficient Θ_1 by up to 56% whereas Θ_2 remained constant (Fig. 5d). Comment: In the case of the HEDS reaction, what is the step referred to by θ_1 , Grx-SSEtOH formation or Grx-SSG formation? How would this be distinguished? (See next question)"

Reply 1.9: We interpret the decreased reciprocal Dalziel coefficient 1 as the GSSEtOH-dependent formation of Grx-SSG (i.e., Grx-SH + GSSEtOH) in accordance with the scaffold model. Considering the size of HEDS and the distance between E170 and the active site thiolate, HEDS is too small to be

affected by the mutation of E170, whereas GSSEtOH should be affected in a similar way as GSSCys. The latter scenario is supported by the charge and type-dependent data for E170X mutants in Table S1. This interpretation is summarized in the sentence that is cited by the referee (see next reply).

10. p.11, 289-91, "The decrease of the $1/\Phi_1$ values (Fig. 5d) therefore suggests a less efficient interaction between E170X mutants and HEDS-derived GSSEtOH in accordance with the scaffold model and the hypothesis that GSSEtOH formation is catalyzed by ScGrx7." /Comment: There is concern that these kinetics (especially for HEDS) may be peculiar to this form of GRx, precluding broad application of the interpretations. At least one more active form of GRx should be tested in order to validate the proposed catalytic mechanisms.

Reply 1.10: Our additional experiments on HEDS and PfGrx support the model (see reply 1.1).

11. p. 12, ls. 331-332 "Furthermore, the assignment of the glutathione scaffold site could facilitate the pre-diction of disulfide substrate preferences of uncharacterized members of the thioredoxin family." / Comment: How can a scaffold site be identified by a single residue? What other residues are important to this interaction? (see next comment where multiple residues are suggested for the GSH activator site besides K105).

12. p. 13, ls. 350-352, "Regarding additional residues of ScGrx7 that could form an interaction site for GSH, we suggest Asp144 and Glu147 in helix 3 as well as the hydroxyl group of conserved Tyr110/ Comment: Although no mutagenesis studies are provided to support these interpretations, still this discussion is a better attempt to describe an activation motif than the assignment of a single residue as the scaffold site.

Reply 1.11 and 1.12: We agree that we should have added the precise residue candidates at the scaffold site. These are now mentioned in the revised version of the manuscript ("Based on existing glutaredoxin structures, the previously published model of ScGrx7¹¹ and our kinetic data, we can now assign the glutathione scaffold site for ScGrx7. The site includes Glu170 as well as candidate residues Gln143 (r2) in helix 3, Arg152 (r7) and Thr154-Pro156 in a loop preceding strand 3, and Gly167-Thr169 (r4-6) in helix 4 (Fig. 1c, Fig. 6a)."). We now included supporting data for our model for the homologous residues of the non-related enzyme PfGrx. Preliminary data on Tyr110 at the activator site of ScGrx7 also support our model but are not included in the revised manuscript because a mapping of further scaffold/activator candidate residues requires future studies and goes beyond the scope of this work.

13. Methods, p. 20, l. 544, "All assays were started by addition of enzyme." Comment: Does this mean that substrates were pre-incubated? If so, for how long? It is important to describe the time allotted for temperature equilibration, especially for the HEDS reaction where non-enzymatic formation of GSSEtOH would occur and affect the interpretation of the reaction mechanism.

Reply 1.13: All HEDS assays were performed with a preincubation period of 2 min as described previously in the indicated references 11, 12 and 18. We added the information for clarity as requested ("HEDS, GSH and NADPH were preincubated in assay buffer for 2 minutes before GR was added and a baseline was recorded for 30 seconds"). Please note that longer preincubation has a surprisingly little effect on ScGrx7 catalysis as published previously (Fig. 5 in Begas et al 2015 Chem Science). This finding also supports the enzymatic conversion of HEDS yielding Grx-SSEtOH as an intermediate (see also reply 6.1).

14. Re: supplementary figures/ What were the criteria for omitting the values shown in parentheses from the regression analyses? How were the lines fit - linear regression of the double reciprocal values, or non-linear fit to rectangular hyperbolic relationships? These details should be included.

Reply 1.14: The apparent k_{cat} and K_m values in the tables and the secondary plots are based on the non-linear fits of the direct Michaelis-Menten plots. In addition, we always analyze our data using linear fits for L.B. plots as well as Hanes- and Eadie-Hofstee plots (not shown) in order to identify kinetic patterns and to compare the resulting apparent k_{cat} and K_m values (see also Deponte et al. 2007 JBC). The advantage of the comparison of the kinetic constants from the four different plots is that their regression analyses have different statistical susceptibilities towards incorrect substrate concentrations or enzyme concentrations (as outlined in ref. 49) and that pipetting errors for substrates or enzymes can be easily detected. If the apparent k_{cat} and K_m values deviate significantly between the four different plots, outliers are identified by systematically omitting single data points from all four regression analyses. A data point was only omitted in the final graph when the apparent k_{cat} and K_m values converged for all four plots after removal of the outlier. These and other details were added to the supplementary figure legends as requested.

15. Re: Supplementary Tables/ There should be more complete legends for the supplementary tables. In Supplementary Tables 2-6, how are the errors determined? These errors should be included in the values of "percent relative to wild type," which should be defined in the legend. Importantly, significant differences should be indicated.

Reply 1.15: Statistical analyses with P-values for all apparent k_{cat} and K_m values were added as requested in a separate Supplementary Table 11 (10 pages). This information is mentioned in the table legends together with the requested information on the determination of the error bars and the definition of the percentages. We did not add the \pm errors to the percentages in order to maintain the readability of the tables.

Reviewer #2 (Remarks to the Author):

The paper entitled "Glutaredoxin catalysis requires two distinct glutathione interaction sites" by Patricia Begas and collaborator describes a thorough kinetic study of a yeast glutaredoxin, namely GRX7. They have analyzed under steady-state conditions the kinetic parameters of an intact recombinant protein and of a certain number of variants mutated for two residues (Lys105 and Glu170) with two different substrates thought to be representative of glutathionylated substrates (L-cysteine-glutathione disulfide, GSSCys) or of non-glutathione substrates (bis(2-hydroxyethyl)disulfide, HED). The aim was to experimentally validate or invalidate two previously proposed hypothetical models for glutaredoxin catalysis, which basically assumed the existence of two distinct protein areas, one for the recognition of the disulfide substrate (scaffold site) and one for the recognition of the reducing glutathione (activator site). These two residues have been selected based on a structural amino acid alignment and on their conservation in glutaredoxins which exhibit thiol disulfide exchange reactions compared to those which do not generally exhibit GSH-dependent activity at least with HED. The enzymatic experiments seem to have been very carefully conducted. Considering the very specialized expertise needed to understand the data, they are quite clearly presented with appropriate statistics.

The interpretation of the kinetic data made by the authors suggests that there are indeed two successive steps for recognizing glutathione moieties involving at least two different residues, Glu170 being part of the scaffold site whereas Lys105 is rather part of the activator site. Other residues have been proposed to participate to these respective sites, but there is no associated experimental demonstration.

If these observations turn to be true for other residues and other enzymes, this would constitute a major advance in our understanding of the biochemical properties of the different Grx subclasses. In

the absence of generalization, this might be my major criticism/concern. Does this model really explain differences between what has been defined as "active and inactive" glutaredoxins.

Reply 2.1: We thank the referee for his careful review and constructive criticism. We now included additional kinetic data on the homologous residues in the non-related enzyme PfGrx from malaria parasites. We also analyzed the catalytically 'inactive' monothiol GRXS15 from Arabidopsis using reduced and oxidized roGFP2 as a substrate. These experiments indicate that GRXS15 reacts with GSSR but not with GSH as predicted. The additional data sets are included as Fig. 11-17 and Supplementary Tables 6-8 and are presented and discussed in the corresponding paragraphs of the revised manuscript. Furthermore, we performed a preliminary analysis on Tyr110 as a third residue in ScGrx7. The data are in accordance with our model but were not included in the manuscript because we need more replicates and mutants to confirm the findings and a mapping of further scaffold/activator candidate residues goes beyond the scope of this work (see also reply 2.5.). As requested by another referee, we also listed P-values for all apparent k_{cat} and K_m values in a supplementary table and specified some figure legends in the supplements.

I have a relatively few minor other concerns detailed below:

- In the abstract (two times) and at many places, the authors use the definition "enzymatically active and inactive glutaredoxins". This is an incorrect shortcut because as rightly defined once in the introduction, this definition essentially applies to the glutathione-dependent reactions and even more specifically with HED as a substrate. These presumed inactive glutaredoxins may in fact have specific substrates and reducing pathways as shown or suggested by a few studies.

Reply 2.2: The referee is right about the terminology. The term "enzymatically active" indicates that the glutaredoxin is active in standard assays, e.g., with HEDS or GSSCys, whereas "inactive" isoforms might have special substrates. We corrected the abstract and introduction accordingly. Furthermore, we added the following information to the introduction for clarity and to avoid repetitive verbose sentences throughout the manuscript:

Abstract: "The exact structure-function relationships and mechanistic differences among glutaredoxins that are active or inactive in standard enzyme assays have so far remained elusive despite numerous kinetic and structural studies. (...) We propose that the requirement of two distinct glutathione interaction sites for the efficient reduction of glutathionylated disulfide substrates explains the deviating structure-function relationships, activities and substrate preferences of different glutaredoxin subfamilies as well as thioredoxins."

Introduction: "Based on such standard assays, different isoforms are hereinafter referred to as "enzymatically active or inactive glutaredoxins" for the sake of simplicity (without excluding the possibility that inactive isoforms might actually catalyze other reactions with specialized substrates in vivo)."

- Lines 114-118: "In accordance with the glutathione activator hypothesis, only positively charged amino acids replace Lys105 in enzymatically active glutaredoxins, whereas in inactive isoforms and other members of the thioredoxin superfamily the activator candidate is often separated from the catalytic cysteine residue or is replaced by uncharged residues^{3,11,12} (Fig. 1c). »

It is indicated that the amino acid alignment has been manually adjusted based on structural overlays and indeed this lysine is very conserved in Grxs and perfectly aligns in both "active and inactive" isoforms. So, does it mean that the Lys is at the right or same position in inactive GRXs although separated from the catalytic Cys by a few residues? This has to be carefully verified from available structures.

Reply 2.3: The backbone atoms of the conserved lysine residue are approximately at the same position in active and inactive Grx. Depending on the isoform and the presence or absence of glutathione in the structures, the side chain conformation of the residue can vary significantly. The presence of the lysine residue in most glutaredoxins becomes understandable considering that the residue also stabilizes the active site thiolate, which is advantageous for the oxidative half-reaction as well as the interaction with Fe/S clusters. Thus, the presence of the conserved lysine residue is a necessary but not a sufficient condition for GSH activation. Other residues such as Tyr110 are also involved in GSH activation (see also 2.5.) or prevent the Grx-GSH interaction. We added this information to the discussion for clarity (e.g., "Please note that Lys105 and homologous residues adopt quite similar positions in the structures of active and inactive glutaredoxins³. Thus, the presence of the basic residue seems to be a necessary but not a sufficient condition for GSH activation.").

- Line 126-127 "...might be sandwiched between the protein and the glutathione moiety of GSH". There should be something wrong here.

Reply 2.4: We are sorry that we do not fully understand what the referee considers to be wrong in the sentence but rewrote the sentence for clarity ("Furthermore, we took into account that the glutathione moiety that originates from the GSSR disulfide substrate might be sandwiched between the protein and the glutathione moiety of GSH approaching the active site (Fig. 1b,d).").

- While the Lys105 is extremely conserved, Glu170 is less conserved although the amino acid sequences used for the alignment have been judiciously chosen. Moreover the proposal that Asp144, Glu147 and Tyr110 could be part of the "scaffold" interaction site with GSH is more speculative in the absence of experimental support and of 3D structure for ScGrx7 and considering that Asp144 and Glu147 are relatively weakly conserved and that Tyr110 is replaced by a Phe in most if not all "inactive" Grxs but this Phe is found also in an important set of "active" Grxs. Considering these observations and in the absence of further kinetic and structural data and since the authors validated their model of "scaffold and activator sites" only for one residue, I would be more cautious with the description of the glutaredoxin active site as two distinct glutathione interaction sites. In fact, this can also be presented as the successive involvement of specific residues during the catalytic act.

Reply 2.5: We agree that Grx-catalysis is an orchestrated process and that some glutaredoxins might share a combination of residues that allows a replacement of Tyr110 or other residues. For example, the moderate conservation of Glu170 at the scaffold site is plausible considering that several residues altogether contribute to the GSSR interaction site as revealed in the numerous structures of glutathionylated glutaredoxins. In order to further substantiate our findings, we now included kinetic data for Lys26 and Asp90 mutants of PfGrx and analyzed the GRXS15-catalyzed reduction and oxidation of roGFP2 by GSH and GSSG. The additional data sets are included as Fig. 11-17 and Supplementary Tables 6-8 and are presented and discussed in the corresponding paragraphs of the revised manuscript. Please note that all data are in accordance with our model and support our generalized statements on the catalytic mechanism of Grx.

Regarding residue Tyr110 we also performed a preliminary mutational analysis of ScGrx7. The detected effects are similar to the effects for the Lys105 mutants suggesting a contribution of Tyr110 to the activator site of ScGrx7 in accordance with our model. The confidential preliminary results on Tyr110 are not included in the manuscript because we need more replicates and a mapping of further scaffold/activator candidate residues such as Tyr110, Asp144 and Glu147 goes beyond the scope of this work. Please take into account that the manuscript already contains numerous data sets and that a thorough site mapping would require several additional time-consuming experiments.

Other comments:

- Line 353: "Regarding additional residues of ScGrx7 that could form an interaction site for GSH, we suggest Asp144 and Glu147 in helix 3 as well as the hydroxyl group of conserved Tyr110: First, helix 3 and Tyr110 are both replaced in enzymatically inactive glutaredoxins^{11,12} (Fig. 1c). Second, the four residues stick out on the protein surface, are on top of the scaffold site".

I guess the number four refers to these three residues and to Glu170 but this is not clear as it presently states.

Reply 2.6: We thank the referee for pointing out this imprecise statement and included Lys105 (not Glu170) in the revised version: "Second, Lys105, Tyr110, Asp144 and Glu147 all stick out on the protein surface..."

-Fig 1b: According to the legend and unless I am missing something, I would have expected to see a GSSR and not a GSSG in the "glutathione scaffold model" part.

Reply 2.7: We thank the referee for pointing out this source of ambiguity. The referee is right that the scaffold model depicts the interaction with the glutathione moiety that is derived from the GSSR substrate. Unless there is a conformational change, this interaction remains the same for the transition state with GSSG. In order to highlight the mechanistic difference between the scaffold and the activator model without showing the exhaustive reaction sequences from refs. 3 and 12, we decided to depict the second transition state with GSSG for the activator and the scaffold model and to color-code the glutathione moieties (GS from GSSR in red and GS from GSH in blue). The absent information on the color-coding and the GSSG transition state was probably confusing and is now included in the figure legend for clarity (" The models distinguish protein-substrate interactions with the glutathione moieties of GSSR in red and GSH in blue^{3,12}. Only the transition state yielding GSSG is shown for the sake of simplicity.")

Reviewer #3 (Remarks to the Author):

The manuscript 'Glutaredoxin catalysis requires two distinct glutathione interaction sites' by M. Deponte and co-workers addresses the basis for the reduction of the model substrates HEDS and Cys-S-SG by glutaredoxins with one molecule of GSH. The authors constructed two mutants and analysed the steady state kinetics of these proteins to an unparalleled extend. Based on these analysis, the authors concluded two distinct glutathione interactions sites. First, the site that interacts with the glutathione moiety of glutathionylated disulfide substrates, and secondly, the site that interacts with glutathione as the reducing agent of the intermediate mixed disulfide between the glutaredoxin and the first glutathione.

Glutaredoxins, the structurally and functionally most diverse group of the thioredoxin fold family, possess fundamental roles in metabolic electron supply, cell signalling, and iron metabolism. Many of the proteins are essential for the development or viability of the respective organisms. Hence, the topic of the manuscript may be of interest to a general audience. In fact, I am convinced that this study will also be appreciated by a wide audience and may thus be suitable for publication in Nature Communications.

Based on their in-depth and in many aspects innovative kinetic analysis, the authors in deed provide the first experimental evidence for two distinct glutathione (moiety) interaction sites for both the ping and the pong reactions of the glutaredoxin reactions cycle.

While the first binding site was well established before, primarily based on solved structures of GSH-Grx complexes, the existence of the second interaction site is the primary novelty of this study.

These findings solve the paradox of glutathione binding despite the lack of any reactivity, as in CGFS-type glutaredoxins. The conclusions drawn are comprehensible, the discussion is balanced and appropriately critical. In fact, a brief structural comparison of different structures of GSH-glutaredoxin complexes (E. c. Grx3 and H.s. Grx2) confirmed that GSH in FeS-Grx/GSH complexes and in mixed disulfides (subsequent to the first half-reaction) occupy the same binding site.

Undoubtedly, these results represent a major step in our understanding of the diverse functions of glutaredoxins in redox biochemistry and metal homeostasis.

Suggested improvements:

In their study, the authors demonstrated the importance of E170 (S.c. Grx7 numbering) for the first interaction site. The authors also discuss the potential involvement of other residues and side chains for this interaction. It would be extremely valuable if the authors would put some more of these residues to the test, especially the hydroxyl group of the active site Y110.

Reply 3.1: We thank the referee for his highly encouraging comments and suggestions. We had the choice to either add more data on ScGrx7 to the manuscript or to test and substantiate our model with other Grx. As outlined in our comments to referees #1 and #2 we decided to focus on the latter aspect and (i) analyzed GRXS15 from Arabidopsis using reduced and oxidized roGFP2 as a substrate and (ii) studied the homologous residues Lys26 and Asp90 in the non-related enzyme PfGrx from malaria parasites. The additional data sets are included as Fig. 11-17 and Supplementary Tables 6-8 and are presented and discussed in the corresponding paragraphs of the revised manuscript. We also performed a preliminary analysis of Tyr110 as a third residue in ScGrx7 as suggested by referee #3. The confidential data support our model suggesting a similar role for Lys105 and Tyr110 but we need more replicates and mutants to confirm the findings. As requested by referee #1, we also listed P-values for all apparent k_{cat} and K_m values in a supplementary table and specified some figure legends in the supplements.

It would be helpful if the authors also include a brief discussion on the impact of their findings for our understanding of iron-sulfur ligation by glutaredoxins in concert with glutathione

Reply 3.2: We added a paragraph on the reaction between AtGrxS15 and roGFP2 to the results and rewrote the discussion, so that differences between the glutathione interaction sites of iron-sulfur cluster-containing glutaredoxins and enzymatically active glutaredoxins are highlighted. The physiological implications are also mentioned now.

Some details on the statistical analysis of the results from the kinetic measurements are requested to be able to better judge the robustness of the conclusions drawn.

Reply 3.3: We now updated several sections on the statistical analyses in the figure legends as requested and also added a supplementary table 11 with P-values for all apparent k_{cat} and K_m values. The P-values were not directly added to the figures to avoid a crowding of symbols that would render the bar charts unreadable.

Reviewer #1 (Remarks to the Author)

Overall, the authors have provided a very effective rebuttal to the issues raised about the initial form of the manuscript, addressing most of the key concerns and improving the report substantially. In particular, the manuscript has been enhanced by the inclusion of data for two other forms of Grx that corroborate the original conclusions based on only one form. The only unsatisfying response is the allusion to preliminary data for mutation of Tyr110, referred to as confidential and beyond the scope of the current manuscript. This is troublesome because the interpretation by the authors is that Tyr110 may contribute to both the scaffold site and the activation site. Thus, it is conceivable that Tyr110 may be part of a conformationally flexible hinge region adjoining the scaffold and activation sites, which reciprocally controls the binding of the glutathionyl moiety of the mixed disulfide and the activation of the GSH. Although other residues are now named as likely contributing to the scaffold site, still data are provided for mutation of only Glu170 and the impact of this mutation is very small.

Reviewer #2 (Remarks to the Author)

In the revised version of this manuscript "Glutaredoxin catalysis requires two distinct glutathione interaction sites" by Begas et al, the authors have now included kinetic data on other representative glutaredoxins: (i) PfGrx, a Plasmodium falciparum dithiol Grx, and two variants similar to those used for ScGRX7 and (ii) GRXS15, an Arabidopsis thaliana monothiol Grx with a CGFS signature. My previous concern on this aspect has thus been addressed. My second major concern (which was also pointed by reviewer1) was that only one residue at each putative GSH interaction site has been mutated to validate the proposed model. Although I understand that this would represent an important amount of work, it is still regrettable that there is no experimental evidence for other residues. Since the major novelty of this work is the characterization of the activator site, having the results for Tyr110 mutation integrated could considerably strengthen the conclusions.

Reviewer #3 (Remarks to the Author)

All of our previous concerns have been addressed appropriately. I am convinced that the manuscript will be highly appreciated by the field and beyond. Publication is recommended.

Reviewers' comments:

Reviewer #1 (Remarks to the Author):

Overall, the authors have provided a very effective rebuttal to the issues raised about the initial form of the manuscript, addressing most of the key concerns and improving the report substantially. In particular, the manuscript has been enhanced by the inclusion of data for two other forms of Grx that corroborate the original conclusions based on only one form. The only unsatisfying response is the allusion to preliminary data for mutation of Tyr110, referred to as confidential and beyond the scope of the current manuscript. This is troublesome because the interpretation by the authors is that Tyr110 may contribute to both the scaffold site and the activation site. Thus, it is conceivable that Tyr110 may be part of a conformationally flexible hinge region adjoining the scaffold and activation sites, which reciprocally controls the binding of the glutathionyl moiety of the mixed disulfide and the activation of the GSH. Although other residues are now named as likely contributing to the scaffold site, still data are provided for mutation of only Glu170 and the impact of this mutation is very small.

Reviewer #2 (Remarks to the Author):

In the revised version of this manuscript "Glutaredoxin catalysis requires two distinct glutathione interaction sites" by Begas et al, the authors have now included kinetic data on other representative glutaredoxins: (i) PfGrx, a Plasmodium falciparum dithiol Grx, and two variants similar to those used for ScGRX7 and (ii) GRXS15, an Arabidopsis thaliana monothiol Grx with a CGFS signature. My previous concern on this aspect has thus been addressed. My second major concern (which was also pointed by reviewer1) was that only one residue at each putative GSH interaction site has been mutated to validate the proposed model. Although I understand that this would represent an important amount of work, it is still regrettable that there is no experimental evidence for other residues. Since the major novelty of this work is the characterization of the activator site, having the results for Tyr110 mutation integrated could considerably strengthen the conclusions.

Reviewer #3 (Remarks to the Author):

All of our previous concerns have been addressed appropriately. I am convinced that the manuscript will be highly appreciated by the field and beyond. Publication is recommended.

Reply: We now included the data set on Tyr110 as requested (Supplementary Figures 18 and 19) and added a corresponding paragraph to the results section. We hope that the manuscript can now be published in its current form without additional experiments.

Please note that testing other scaffold site candidates will either have similar minor effects as observed for Glu170 (because the oxidative half-reaction is not rate-limiting) or result in mixed secondary effects that will complicate the data interpretation (see also line 132-139). While we share the interest of Reviewers #1 and #2 for further mechanistic and structural insights - e.g., on a potential hinge region, conformational changes and additional residues contributing to the glutathione interaction sites - none of our key findings and statements of this study would be altered by the inclusion of additional kinetic data. Answering all these aspects goes clearly beyond the scope of a single manuscript (which already contains a significant amount of novel data). We will try to address some of the questions in thorough follow-up stories that will require numerous controls for each candidate residue in order to discriminate between shape- and charge-dependent effects on substrate interactions, pKa values and protein conformations.

Reviewer #2 (Remarks to the Author)

In this revised version, the authors have fully answered the concerns raised previously.